# Structural basis of meiotic telomere attachment to the nuclear envelope by MAJIN-TERB2-TERB1

James M. Dunce[1], Amy E. Milburn[1], Manickam Gurusaran[1], Irene da Cruz[2], Lee T. Sen[1], Ricardo Benavente[2] & Owen R. Davies [1]

Meiotic chromosomes undergo rapid prophase movements, which are thought to facilitate the formation of inter-homologue recombination intermediates that underlie synapsis, crossing over and segregation. The meiotic telomere complex (MAJIN, TERB1, TERB2) tethers telomere ends to the nuclear envelope and transmits cytoskeletal forces via the LINC complex to drive these rapid movements. Here, we report the molecular architecture of the meiotic telomere complex through the crystal structure of MAJIN-TERB2, together with light and X-ray scattering studies of wider complexes. The MAJIN-TERB2 2:2 hetero-tetramer binds strongly to DNA and is tethered through long flexible linkers to the inner nuclear membrane and two TRF1-binding 1:1 TERB2-TERB1 complexes. Our complementary structured illumination microscopy studies and biochemical findings reveal a telomere attachment mechanism in which MAJIN-TERB2-TERB1 recruits telomere-bound TRF1, which is then displaced during pachytene, allowing MAJIN-TERB2-TERB1 to bind telomeric DNA and form a mature attachment plate.

---

[1] Institute for Cell and Molecular Biosciences, Faculty of Medical Sciences, Newcastle University, Framlington Place, Newcastle upon Tyne NE2 4HH, UK. [2] Department of Cell and Developmental Biology, Biocenter, University of Würzburg, D-97074 Würzburg, Germany. These authors contributed equally: James M. Dunce, Amy E. Milburn, Manickam Gurusaran, Irene da Cruz. Correspondence and requests for materials should be addressed to R.B. (email: benavente@biozentrum.uni-wuerzburg.de) or to O.R.D. (email: owen.davies@newcastle.ac.uk)

The establishment of homologous chromosome pairs during meiotic cell division is achieved through extensive recombination-mediated homology searches to generate a series of recombination intermediates, hence physical linkages, between homologues[1,2]. This challenging task is accomplished through a unique chromosomal choreography during meiotic prophase I[3]. The telomeric ends of meiotic chromosomes become physically tethered to the nuclear envelope, where cytoskeletal forces are transmitted to chromosome ends to drive their rapid movement throughout the nuclear envelope (Fig. 1a)[4,5]. These rapid prophase movements peak during establishment of recombination intermediates in zygotene[6,7], when on the basis of studies in yeast and *C. elegans*[8,9], they are thought to facilitate the homology search by driving interactions between chromosomes and disrupting non-homologous contacts. The resultant recombination intermediates direct assembly of the synaptonemal complex (SC), a supramolecular protein lattice that provides continuous synapsis between homologous chromosome axes[10–12]. The SC provides the necessary three-dimensional framework for recombination resolution and crossover formation[13,14], with its structure terminating at synapsed telomere ends through attachment plates that are physically fused to the inner nuclear membrane[15–18].

In mammals, meiotic telomere attachments are achieved by a meiosis-specific complex of MAJIN, TERB1 and TERB2 (herein referred to as the meiotic telomere complex) that repurposes and integrates the functions of two complexes, LINC and shelterin, which otherwise perform distinct critical roles outwith meiosis. The linker of nucleoskeleton and cytoskeleton (LINC) complex provides the physical connection that mediates cytoskeletal force transduction from the cytoplasm to the nucleus and is formed through a peri-nuclear interaction of SUN and KASH domain proteins[19–21]. KASH proteins cross the outer nuclear membrane to interact with the cytoskeleton[22–24], whilst SUN proteins cross the inner nuclear membrane to interact with nuclear lamin A and emerin[25,26]. Accordingly, LINC complexes are largely immobile

structures that have universal and essential functions in nuclear structure and positioning[19–21]. However, in meiotic prophase I, the nuclear lamina undergoes an extensive reorganisation[27] and LINC complexes become associated with the meiotic telomere complex[28], thus transmitting cytoskeletal forces to meiotic telomere end assemblies that are freely mobile within the plane of the nuclear envelope. These movements are achieved by the partially redundant universally expressed SUN domain proteins SUN1 and SUN2[29–31], and a meiosis-specific KASH domain protein, KASH5, which is essential for meiotic telomere attachments[22,32]. KASH5 interacts with microtubules via dynein–dynactin[22,32], accounting for the autonomous movements of chromosome ends along stationary microtubule tracts that have been visualised in meiotic prophase mouse spermatocytes[6,7].

The shelterin complex, which includes TRF homology domain proteins TRF1 and TRF2, TIN2, TPP1 and POT1, protects telomeric ends from DNA damage response activation and recruits telomerase to regulate telomere length[33–35]. TRF1 and TRF2 form structural homodimers through their TRF homology domains[36], which mediate direct interactions with TIN2, and preferentially bind telomeric double-stranded DNA through C-terminal MYB domains[37]. TIN2 recruits telomerase-binding protein TPP1, which in turn recruits telomeric single-stranded DNA-binding protein POT1[38,39]. Short peptide motifs of TIN2 and related proteins bind to sites near the dimeric cleft of TRF proteins, providing two possible binding sites per TRF dimer[40]. However, the reconstituted shelterin core of TRF2-TIN2-TPP1–POT1 has a stoichiometry of 2:1:1:1 in vitro[41], suggesting that binding events are limited by steric hindrance within the wider structural assembly. Shelterin provides a means of telomere recognition for the meiotic telomere complex, which physically connects it to the LINC complex, thereby integrating their functions to achieve cytoskeletal attachments of meiotic telomere ends across the nuclear envelope.

Meiotic telomere complex components MAJIN, TERB1 and TERB2 are essential for fertility; their individual disruption in

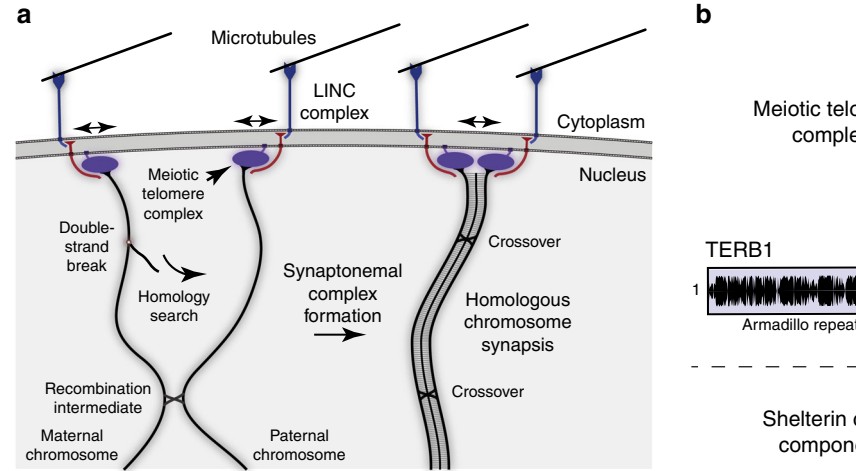

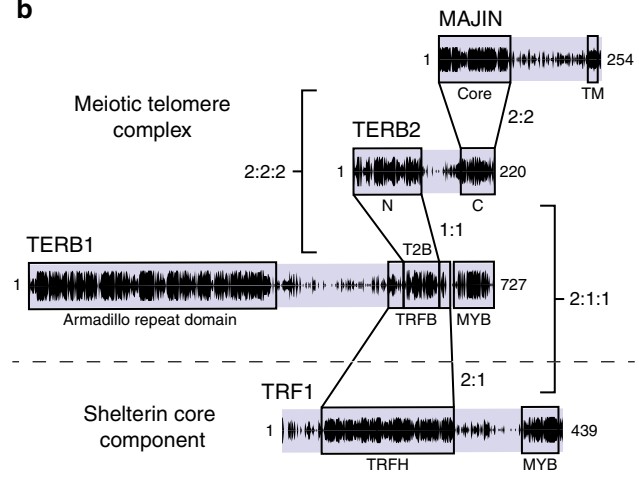

**Fig. 1** Rapid prophase chromosomal movements through meiotic telomere-nuclear envelope tethering. **a** Meiotic chromosome telomere ends are tethered to the nuclear envelope by the meiotic telomere complex, enabling the transmission of cytoskeletal forces from microtubules to chromosome ends through LINC complexes. This produces rapid prophase chromosomal movements that are thought to facilitate recombination-mediated homology searches to generate inter-homologue recombination intermediates. The resulting pairings mature into synapsis through synaptonemal complex assembly, leading to crossover formation and the generation of single end attachment plates between synapsed telomere ends and the nuclear envelope. **b** Schematic of the domain structure and sequence conservation of human meiotic telomere complex proteins MAJIN, TERB1 and TERB2, and shelterin component TRF1. Amino acid conservation is illustrated as black plots in which the plot height represents the per residue conservation score. Sequences are annotated with their domain structure, interacting regions and oligomer states based on previous reports[42–45] and the results presented herein. MAJIN contains an N-terminal globular core (amino acids 1–112) that interacts with the TERB2 C-terminus (amino acids 168–220). The TERB2 N-terminus (amino acids 1–107) interacts with the TERB2-binding (T2B) site of TERB1 (amino acids 585–642), which is flanked by a wider TRF1-binding (TRFB) site (amino acids 561–658) that interacts with the TRF homology (TRFH) domain of TRF1 (amino acids 62–268). MAJIN contains a C-terminal transmembrane (TM) helix, whilst TERB1 and TRF1 both contain C-terminal DNA-binding MYB domains

mice leads to meiotic arrest with failure of telomere attachments and chromosomal movements, failure of DNA double-strand break repair and impaired synapsis[28,42]. Their binary interactions have been mapped by yeast two-hybrid and in vivo interaction and deletion studies[28,42]. MAJIN provides the inner nuclear membrane attachment through a transmembrane helix at its C-terminus[42], whilst TERB1 is proposed to interact with SUN1 and cohesin components through its N-terminal armadillo repeat domain and C-terminal MYB domain, respectively (Fig. 1b)[28]. MAJIN and TERB1 are physically linked by TERB2, which binds MAJIN through its C-terminus and TERB1 through its N-terminus[42,43]. TERB1 also interacts directly with shelterin component TRF1 through a region flanking its TERB2-binding site, including a peptide interaction that mimics TIN2 binding to the TRF1 dimeric cleft[28,40,44,45]. Nevertheless, an interaction of the meiotic telomere complex with TRF1 appears to be transient in the cell. MAJIN–TERB2–TERB1 assembles on the nuclear envelope and undergoes TRF1-dependent recruitment of telomere ends during leptotene and zygotene;[42,43] CDK activity then triggers the displacement of TRF1 to flanking regions in pachytene, with MAJIN–TERB2–TERB1 remaining associated with telomere ends[42]. However, we hitherto lack the structural information regarding the meiotic telomere complex necessary to understand the molecular events underlying meiotic telomere attachment.

Here, we report the crystal structure of the MAJIN–TERB2 complex, revealing a 2:2 heterotetramer in which two TERB2 chains wrap around a core MAJIN globular dimer. This structure undergoes direct interaction with DNA and scaffolds assembly of the full 2:2:2 MAJIN–TERB2–TERB1 meiotic telomere complex, which can recruit two TRF1 dimers. Together, our data lead to a molecular model, in which a hierarchical series of binding events achieve meiotic telomere attachment through TRF1-mediated loading of telomeric DNA to the meiotic telomere complex.

## Results

**Crystal structure of MAJIN–TERB2.** The N-terminal structural core of MAJIN (amino acids 1–112) was recombinantly co-expressed with the C-terminus of TERB2 (amino acids 168–220), yielding a co-purifying equimolar complex (Fig. 1b and Supplementary Fig. 1). The presence of both proteins was essential for their stability, suggesting that the MAJIN–TERB2 complex is constitutive. We solved the X-ray crystal structure of MAJIN–TERB2 at 2.90 Å resolution through single-wavelength anomalous diffraction of a selenomethionine derivative (Table 1 and Supplementary Fig. 2a). Using this as a model for molecular replacement, we proceeded to solve the structure of a truncated complex (MAJIN 1–106 and TERB2 168–207) in an alternative crystal form at 1.85 Å resolution (Fig. 2, Table 1 and Supplementary Figs. 2b–d and 3). Both structures are essentially the same (Supplementary Fig. 2c, d) and the subsequent discussion is based on the latter higher resolution crystal form. The MAJIN–TERB2 structure demonstrates a 2:2 heterotetrameric assembly, in which two TERB2 chains wrap around the surface of a core globular MAJIN dimer (Fig. 2a, b). Size-exclusion chromatography multi-angle light scattering (SEC-MALS) confirmed a 2:2 stoichiometry in solution (Fig. 3a). Size exclusion chromatography small-angle X-ray scattering (SEC-SAXS) revealed scattering curves and ab initio envelopes that closely match the heterotetrameric crystal structure for the truncated construct, with the wider complex demonstrating an elongation consistent with its additional sequence being largely unstructured in solution (Fig. 3b–d, Supplementary Fig. 4a–c and Supplementary Table 1). Thus, the 2:2 oligomer observed in the crystal structure corresponds to the solution state of the MAJIN–TERB2 complex.

The MAJIN protomers adopt a β-grasp fold, in which a β-sheet grasps around a core α-helix in a β(2)-α-β(3) configuration (Fig. 2c–e and Supplementary Fig. 3a, b). The five-stranded

**Table 1 Data collection, phasing and refinement statistics**

|  | MAJIN$_{Core}$-TERB2$_C$ selenomethionine derivative (PDB 6GNX) | MAJIN$_{Core}$-TERB2$_C$ Native (truncated) (PDB 6GNY) |
|---|---|---|
| *Data collection* |  |  |
| Space group | P3$_2$21 | C222$_1$ |
| Cell dimensions |  |  |
| $a$, $b$, $c$ (Å) | 59.88 59.88 159.93 | 59.97 88.39 111.67 |
| $\alpha$, $\beta$, $\gamma$ (°) | 90 90 120 | 90 90 90 |
| Wavelength | 0.9159 Å | 0.9763 Å |
| Resolution (Å) | 49.33 – 2.90 (3.08 – 2.90)$^a$ | 45.35-1.85 (1.89-1.85)$^a$ |
| $R_{meas}$ | 0.136 (3.015) | 0.039 (1.374) |
| $R_{pim}$ | 0.024 (0.553) | 0.014 (0.504) |
| Completeness (%) | 100.0 (100.0) | 100.0 (100.0) |
| $I/\sigma(I)$ | 28.2 (2.3) | 23.1 (1.5) |
| $CC_{1/2}$ | 1.000 (0.885) | 0.999 (0.744) |
| Redundancy | 56.6 (55.1) | 7.3 (7.3) |
| *Refinement* |  |  |
| Resolution (Å) | 49.33 – 2.90 | 45.35-1.85 |
| No. of reflections | 7865 | 25690 |
| $R_{work}/R_{free}$ | 0.2542/0.3039 | 0.1883/0.2072 |
| No. of atoms | 2197 | 2413 |
| Protein | 2197 | 2286 |
| Ligand/ion | 0 | 8 |
| Water | 0 | 119 |
| *B* factors | 130.46 | 64.58 |
| Protein | 130.46 | 64.72 |
| Ligand/ion | N/A | 130.48 |
| Water | N/A | 57.49 |
| R.m.s deviations |  |  |
| Bond lengths (Å) | 0.003 | 0.003 |
| Bond angles (°) | 0.539 | 0.627 |

$^a$Values in parentheses are for highest-resolution shell

grasping β-sheet contains an insertion between β4 and β7 strands, which forms a two-stranded β-sheet appendage that splays from the base of the structure. TERB2 chains follow a meandering path from their N-termini at the MAJIN dimer interface, passing around the α-helix to complete its closure and providing an additional short β-strand to the grasping β-sheet, and terminate by forming a third β-strand of the MAJIN β-sheet appendage (Fig. 2f, g). This interaction is mediated largely through the insertion of hydrophobic side-chains of TERB2 into pockets on the MAJIN molecular surface, in addition to a salt bridge between MAJIN residue R14 and TERB2 residue D191, and the aforementioned β-sheet interactions. At its N-terminal end, TERB2 residue Y176 contributes to the MAJIN dimer interface that contains a hydrophobic core formed of highly conserved MAJIN residues P64, F73 and Y75 (Fig. 2h and Supplementary Fig. 3c, d). Whilst the 2:2 assembly is stable across a range of concentrations and chemical conditions, with smaller oligomers never observed in vitro, its disruption through MAJIN mutation F73E Y75E yielded a stable 1:1 complex with secondary structure composition matching that of a single MAJIN–TERB2 protomer (Fig. 3a and Supplementary Fig. 4d, e). Thus, each MAJIN–TERB2 protomer folds and retains stability independent of dimerization, raising the question of how dimerization functions in the wider architecture of the meiotic telomere complex?

The C-terminus of the MAJIN core, which emanates from the final β-strand, lies on the upper side of the structure in the midline, and continues as a poorly conserved sequence of ~120 amino acids that terminates as a well-conserved 19 amino acid transmembrane helix (Fig. 1b and Supplementary Fig. 1a). Through SAXS analysis of a complex of C-terminally extended MAJIN that includes the full sequence apart from the transmembrane helix (MAJIN$_{\Delta TM}$; amino acids 1–233), we modelled the conformation of the MAJIN C-termini with respect to the crystal structure (Fig. 3b, c, e, Supplementary Fig. 4c and

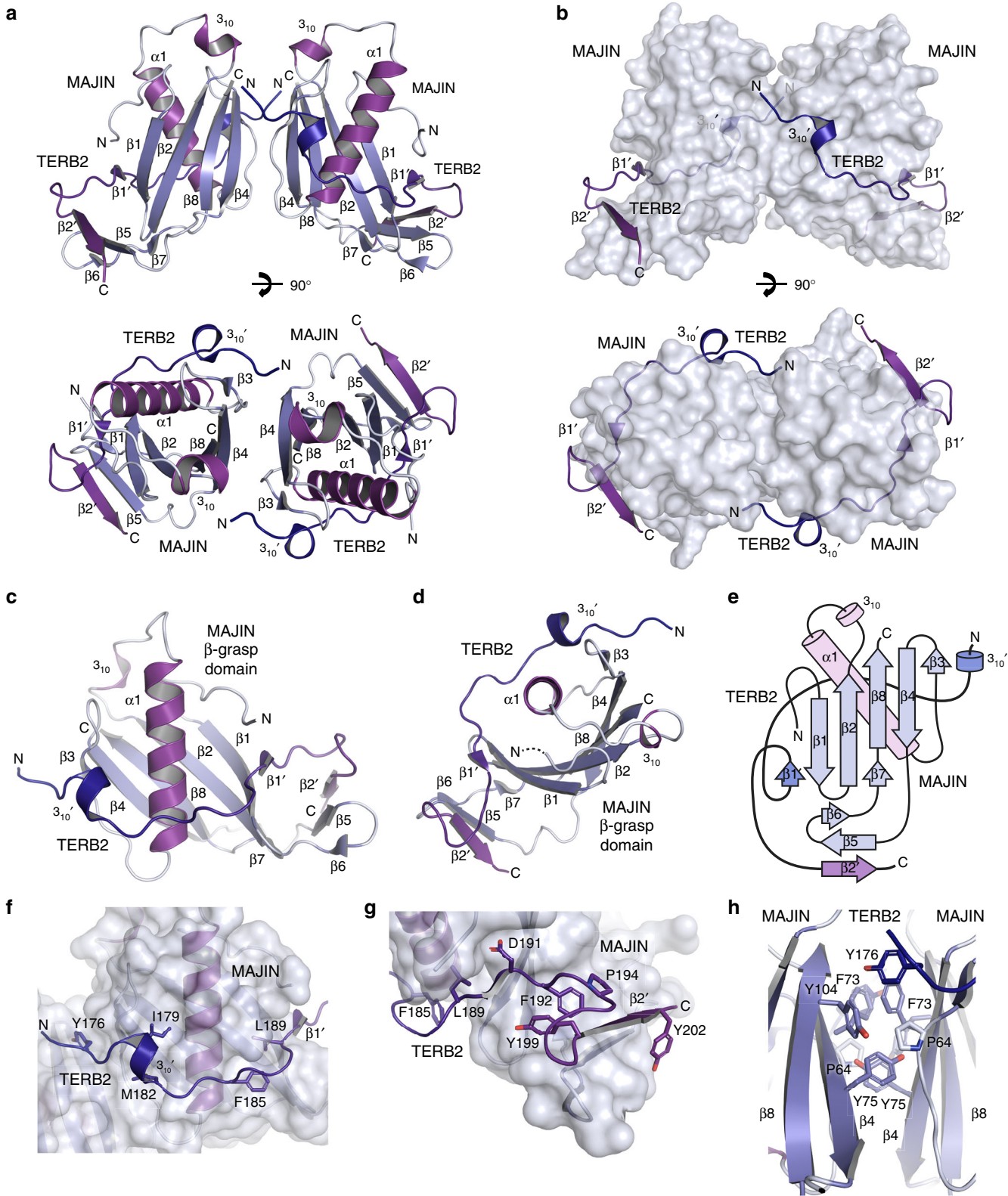

Supplementary Table 1). In agreement with CD analysis (Supplementary Fig. 4d), the SAXS scattering data indicate largely unstructured C-termini that provide flexible linkers of 90 Å in solution (Fig. 3e). Under tension, these linkers could stretch to provide a separation of up to 400 Å between the MAJIN–TERB2 structural core and their transmembrane helices embedded in the inner nuclear membrane.

**DNA-binding by MAJIN–TERB2**. The MAJIN–TERB2 structure shows an extensive basic patch on the surface of each protomer, formed of highly conserved amino acids in the grasping β-sheet of MAJIN (Fig. 4a, b and Supplementary Fig. 3e), suggesting that it may mediate direct DNA-binding through electrostatic interaction with the phosphodiester backbone. Accordingly, analysis of MAJIN$_{Core}$-TERB2$_C$ by electrophoretic mobility shift assay

**Fig. 2** Crystal structure of MAJIN$_{Core}$-TERB2$_C$. **a, b** Crystal structure of MAJIN$_{Core}$-TERB2$_C$ (sequences truncated to amino acids 1–106 and 168–207) solved in C222$_1$ spacegroup at 1.85 Å resolution, demonstrating a 2:2 heterotetrameric complex shown as (**a**) cartoon representation and (**b**) molecular surface of MAJIN with cartoon representation of TERB2 chains. MAJIN forms a central globular dimer that is encircled by two TERB2 chains. **c, d** Cartoon representations and (**e**) topology diagram of a single MAJIN–TERB2 protomer. MAJIN adopts a β-grasp fold in which a β-sheet grasps around a central α-helix in a β(2)-α-β(3) configuration, with an additional two-stranded β-sheet appendage inserted between β4 and β7. The TERB2 chain wraps around the exposed surface of the α-helix to complete its closure such that it becomes fully encircled by TERB2 and the grasping β-sheet. **f, g** The MAJIN–TERB2 interface spans an area of approximately 1300 Å$^2$ for each protomer. **f** The N-terminal end of TERB2$_C$ wraps around the central MAJIN α-helix with aromatic and hydrophobic residues packing into pockets on the MAJIN surface. **g** The C-terminal end of TERB2$_C$ provides a short β-strand extension to the grasping β-sheet, a salt bridge between D191 (TERB2) and R14 (MAJIN), a structural wedge (formed by F192, P194 and Y199) between the grasping and appendage β-sheets of MAJIN, and a β-strand extension to the β-sheet appendage. **h** The MAJIN–TERB2 dimerization interface contains approximately 650 Å$^2$ of buried surface area, formed of largely aromatic and hydrophobic interactions between P64, F73, Y75 and Y104 (MAJIN), and Y176 (TERB2)

(EMSA) confirmed a strong interaction with double-stranded DNA that shows no overt sequence specificity for telomere repeats and is retained for single-stranded DNA (Fig. 4c and Supplementary Fig. 4f). We estimated the apparent $K_D$ at 0.55 μM (random sequence double-stranded DNA) through quantification of EMSAs performed using 25 nM FAM-labelled DNA (Fig. 4d), with similar affinities determined for telomeric and single-stranded DNA substrates (Supplementary Fig. 4g). This is slightly weaker than DNA-binding of TRF1, which we estimated at 0.10 μM (Supplementary Fig. 5), in keeping with previous studies[46]. MAJIN$_{Core}$-TERB2$_C$ DNA complexes were visualised by electron microscopy (EM), revealing plaques of protein–DNA, frequently forming circular assemblies of MAJIN$_{Core}$-TERB2$_C$ (dependent on MAJIN dimerization) connected together by DNA chains in a beads-on-a-string fashion (Fig. 4e).

The C-terminal end of TERB2 includes five highly conserved basic residues within a seven amino acids stretch (amino acids 214–220); this region is unstructured in the selenomethionine structure (in which they are included) and is essential for DNA-binding by MAJIN–TERB2 (Fig. 4a–d). Similarly, MAJIN linkers include two distinct basic patches located in the half proximal to the structural core (Fig. 4a, b); their inclusion led to an enhancement of DNA-binding by MAJIN–TERB2 to an apparent $K_D$ of 0.12 μM (Fig. 4c, d), matching the DNA-binding affinity of TRF1. The presence of MAJIN linkers also rescued the disruption of DNA-binding upon C-terminal truncation of TERB2, with an apparent $K_D$ of 0.20 μM (Fig. 4c, d). Furthermore, DNA-binding of the 1:1 MAJIN–TERB2 complex formed by mutant F73E Y75E is diminished (Fig. 4c, d), indicating that DNA-binding of the MAJIN basic surfaces on either side of the 2:2 complex is cooperative. These findings are consistent with a single continuous DNA-binding interface encompassing the TERB2 C-termini, MAJIN β-grasp surfaces and MAJIN flexible linkers of both protomers. To test the role of the MAJIN β-grasp surface in DNA-binding, we generated a mutant (K24M K26E R28E K31D R34E R81D; herein referred to as basic surface mutant) that eliminated its basic charge (Supplementary Fig. 3e, f). Surprisingly, this mutant inhibited MAJIN–TERB2 dimerization, producing a 1:1 complex (Fig. 3a). Nevertheless, its complete inhibition of DNA-binding, in comparison with the binding affinity of the F73E Y75E 1:1 complex, confirms the role of the MAJIN β-grasp surface in DNA-binding (Fig. 4c, d). Thus, we propose that telomeric DNA is looped around MAJIN–TERB2 molecules to achieve cooperativity through binding to the basic interfaces of both protomers (Fig. 4f). In this model, looping occurs within MAJIN linkers and so their extension under tension in vivo may facilitate the formation of long telomere loops that extend into the nuclear lamina towards the inner nuclear membrane.

**Structure of the MAJIN–TERB2–TERB1 meiotic telomere complex**. We next sought to determine the molecular architecture

of the wider MAJIN–TERB2–TERB1 meiotic telomere complex. Similar to MAJIN$_{Core}$-TERB2$_C$, we observed co-purification of recombinantly co-expressed complexes between the TERB2 N-terminal end (amino acids 1–107) and its binding site within TERB1 (TERB1$_{T2B}$; amino acids 585–642), and the ternary complex between MAJIN$_{Core}$, full length TERB2 and TERB1$_{T2B}$ (Fig. 5a and Supplementary Fig. 6a, b). SEC-MALS analysis revealed that TERB2$_N$–TERB1$_{T2B}$ is a 1:1 complex, which in accordance with the MAJIN$_{Core}$–TERB2$_C$ 2:2 complex, undergoes dimerization in the ternary MAJIN$_{Core}$–TERB2–TERB1$_{T2B}$ complex to form an equimolar 2:2:2 hetero-hexamer (Fig. 5b and Supplementary Fig. 6c). Through circular dichroism (CD) spectroscopy and SEC-SAXS analysis, we find that the 1:1 TERB2$_N$–TERB1$_{T2B}$ complex adopts a globular mixed α/β-structure (Supplementary Figs. 6d, e and 7). Interestingly, both TERB2$_N$ and TERB1$_{T2B}$ (and the wider TERB1$_{TRFB}$ construct) form large molecular weight aggregates in isolation (Supplementary Fig. 6f), indicating that they depend upon each other for structural stability and so the 1:1 complex may be constitutive. The MAJIN$_{Core}$–TERB2$_C$ and TERB2$_N$–TERB1$_{T2B}$ structures are separated by a largely poorly conserved stretch of ~60 amino acids in the middle of the TERB2 sequence, suggesting the presence of three separate globular domains within the 2:2:2 ternary complex.

We utilised SAXS rigid body and linker modelling, coupled with ab initio modelling to determine the structure and relative positioning of the components within the meiotic telomere complex. The MAJIN$_{Core}$–TERB2$_C$ crystal structure and SAXS ab initio models of TERB2$_N$–TERB1$_{T2B}$ were positioned and TERB2 linkers modelled through fitting to TERB2$_N$–TERB1$_{T2B}$ and MAJIN$_{Core}$–TERB2–TERB1$_{T2B}$ SAXS scattering curves (Fig. 5c, d, Supplementary Fig. 7 and Supplementary Table 1). The scattering data are consistent with a central orientation of the MAJIN$_{Core}$–TERB2$_C$ heterotetramer, with flexible linkers within the middle of TERB2 providing separation from two distinct TERB2$_N$–TERB1$_{T2B}$ 1:1 complexes (Fig. 5c).

To test our in vitro findings, we analysed the location of MAJIN, TERB2 and TERB1 within the telomeric ends of chromosomes spread from zygotene and late pachytene mouse spermatocytes through structured illumination microscopy (SIM) (Fig. 5e, f and Supplementary Figs. 8, 9). In zygotene, we observe MAJIN$_{Core}$ foci (antibodies raised against mouse amino acids 18–30) enclosed by flanking TERB2 at the telomeric ends of SYCP3-stained axial elements (Fig. 5e and Supplementary Fig. 8a). In late pachytene, the MAJIN foci of paired telomeres partially coalesce above thickened lateral element ends, with a TERB2 signal that largely overlaps in the frontal plane but clearly encircles MAJIN staining when viewed from a lateral-top orientation (Fig. 5f and Supplementary Figs. 8b, c and 9a, b). In contrast, TERB2 and TERB1$_{T2B}$ (antibodies raised against full length and amino acids 525–540, respectively) co-localised when viewed in multiple orientations (Fig. 5f and Supplementary Figs. 8d, e and 9c, d); similar co-localisation was observed in

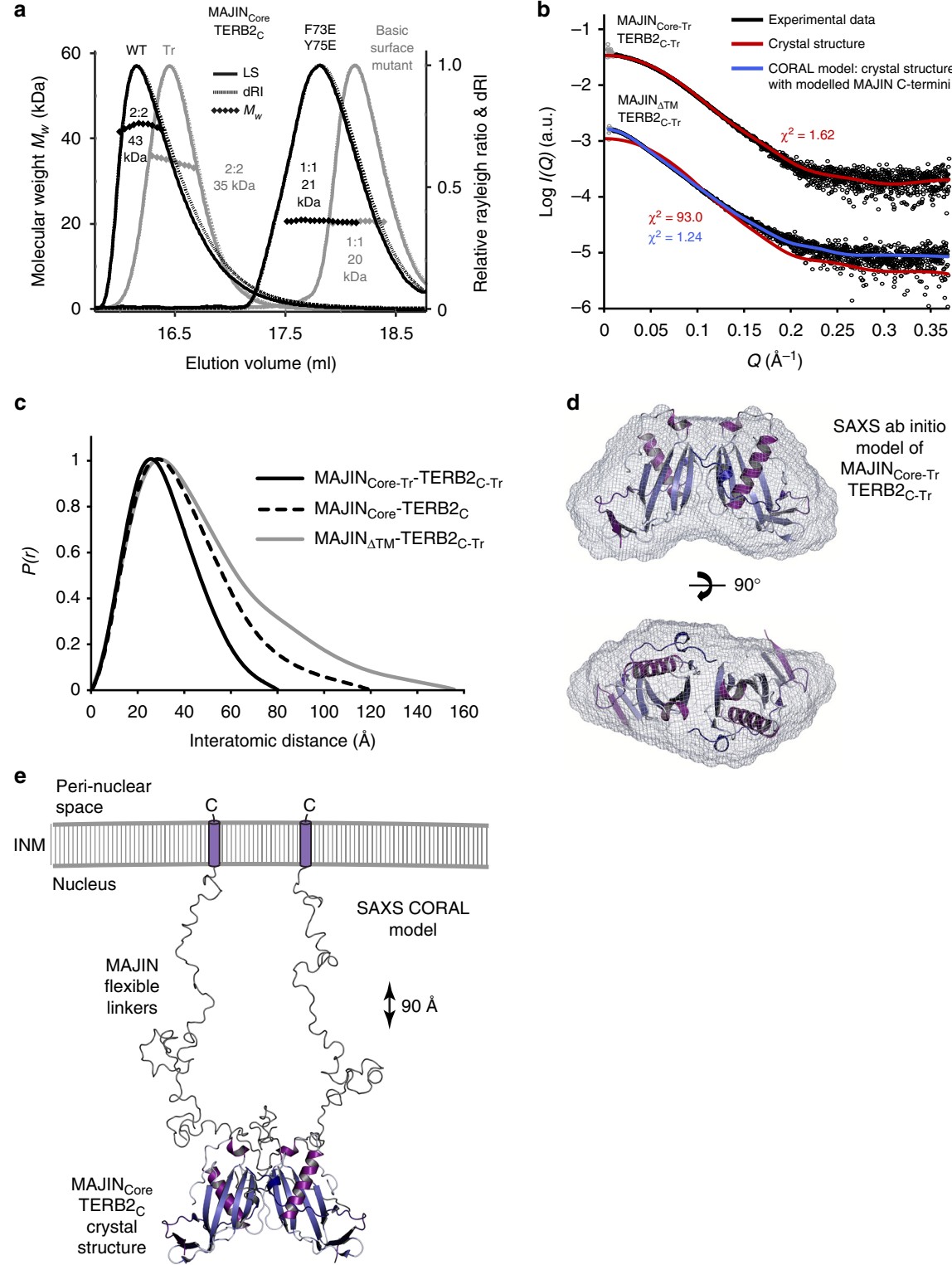

structurally preserved pachytene nuclei (Fig. 5g and Supplementary Fig. 8f). These staining patterns are consistent with our structural model of MAJIN–TERB2–TERB1, in which flexible linkers in the middle of TERB2 are out-stretched, possibly owing to tension forces within the chromosome-attachment plate axis, to provide a physical separation between MAJIN_Core–TERB2_C and TERB2_N–TERB1_T2B globular domains.

We conclude an architecture for the meiotic telomere complex in which a central MAJIN_Core–TERB2_C heterotetramer is tethered on one aspect to the inner nuclear membrane, and on its other aspect to two separated TERB2_N–TERB1_T2B hetero-dimers, through long-flexible linkers within MAJIN and TERB2, respectively (Fig. 5h).

**Structure of the TRF1–TERB1–TERB2 complex.** The shelterin component and telomere-binding protein TRF1 interacts directly with TERB1 through a TRF1-binding domain (TERB1_TRFB;

**Fig. 3** Solution structure of MAJIN–TERB2. **a** SEC-MALS analysis; light scattering (LS) and differential refractive index (dRI) profiles are overlaid, with fitted molecular weights ($Mw$) plotted as diamonds across elution peaks. Wild type (black, left) and truncated (grey, left) MAJIN$_{Core}$-TERB2$_C$ complexes (WT: 1–112, 168–220; Tr: 1–106, 168–207) form 2:2 hetero-tetramers of 43 kDa and 35 kDa, respectively (theoretical 2:2–40 kDa and 35 kDa). MAJIN mutations F73E Y75E (black, right), and K24M K26E R28E K31D R34E R81D (basic surface mutant; grey, right) block dimerization of MAJIN$_{Core}$-TERB2$_C$, leaving 1:1 complexes of 21 kDa and 20 kDa, respectively (theoretical 1:1–20 kDa). **b**–**e** SEC-SAXS analysis. **b** SAXS scattering curves of MAJIN$_{Core-Tr}$–TERB2$_{C-Tr}$ and MAJIN$_{\Delta TM}$–TERB2$_{C-Tr}$ (1–233, 168–207) overlaid with theoretical scattering curves of the crystal structure (red; $\chi^2 = 1.62$ and 93.0, respectively) and a CORAL model of the crystal structure with modelled MAJIN C-termini (blue; $\chi^2 = 1.24$ for MAJIN$_{\Delta TM}$-TERB2$_{C-Tr}$). The scattering data points below the minimum $Q$-value of the Guinier analysis are shown in grey. **c** SAXS $P(r)$ distributions of MAJIN$_{Core-Tr}$-TERB2$_{C-Tr}$ (black, solid), MAJIN$_{Core}$–TERB2$_C$ (black, dashed) and MAJIN$_{\Delta TM}$–TERB2$_{C-Tr}$ (grey), showing maximum dimensions of 80 Å, 120 Å and 155 Å, respectively. Their real space $Rg$ values of 24 Å, 32 Å and 39 Å closely match their Guinier $Rg$ values of 24 Å, 30 Å and 37 Å (Supplementary Fig. 4a–c), respectively. **d** SAXS ab initio model of MAJIN$_{Core-Tr}$-TERB2$_{C-Tr}$. A filtered averaged model was generated from 30 independent DAMMIF runs, imposing P2 symmetry, with NSD = 0.904 (± 0.115) and reference model $\chi^2 = 1.59$, and is displayed as a molecular envelope with the docked crystal structure. **e** SAXS CORAL model of MAJIN$_{\Delta TM}$–TERB2$_{C-Tr}$ in which MAJIN C-termini were modelled onto the MAJIN$_{Core}$-TERB2$_C$ crystal structure up to residue 233 through fitting to experimental SAXS data ($\chi^2 = 1.24$). The model is displayed with a cartoon of C-terminal transmembrane helices inserted in the inner nuclear membrane (INM), highlighting how unstretched MAJIN C-terminal linkers may provide a separation of approximately 90 Å between the nuclear envelope and MAJIN–TERB2 core

amino acids 561–658) that includes and flanks its TERB2-binding site (Supplementary Fig. 10). We achieved co-purification of the TRF homology domain of TRF1 (amino acids 62–268) with TERB1$_{TRFB}$, and of a ternary complex between TRF1$_{TRFH}$, TERB1$_{TRFB}$ and TERB2$_N$, following their recombinant co-expression (Fig. 6a and Supplementary Figs. 10a and 11a–d). SEC-MALS confirmed the well-established dimerization of TRF1$_{TRFH}$ and revealed the recruitment of one TERB1$_{TRFB}$ molecule by the TRF1$_{TRFH}$ dimer to form a 2:1 complex; in accordance with the 1:1 TERB2$_N$–TERB1$_{T2B}$ interaction, the ternary TRF1$_{TRFH}$–TERB1$_{TRFB}$–TERB2$_N$ complex formed a 2:1:1 heterotetramer (Fig. 6b).

We next determined the molecular architecture of the TRF1$_{TRFH}$–TERB1$_{TRFB}$–TERB2$_N$ complex through multi-phase SEC-SAXS ab initio modelling (Fig. 6c, d, Supplementary Fig. 11e–h and Supplementary Table 1). The resultant model demonstrates a V-shaped molecular envelope for TRF1$_{TRFH}$ that closely matches its known crystal structure, with TERB1$_{TRFB}$ bound at its cleft (Fig. 6d), in keeping with the recently reported crystal structures of TRF1$_{TRFH}$ bound to TERB1$_{TBM}$ (amino acids 642–658), a peptide mapping to the C-terminus of TERB1$_{TRFB}$[44,45]. The TERB2$_N$ molecular envelope is closely associated with TERB1$_{TRFB}$ and is distal to the TRF1$_{TRFH}$-binding cleft. We conclude a model for TRF1$_{TRFH}$–TERB1$_{TRFB}$–TERB2$_N$, in which a TRF1$_{TRFH}$ dimer binds to a TERB1$_{TRFB}$–TERB2$_N$ 1:1 complex through a single TERB1-binding site within its dimerization cleft (Fig. 6e).

**TRF1 recruitment by MAJIN–TERB2–TERB1.** The separate ternary complexes that we have described raise the question of how TRF1 may be recruited to the MAJIN–TERB2–TERB1 meiotic telomere complex? We reconstituted a full meiotic telomere recruitment complex through the addition of TRF1$_{TRFH}$ to purified MAJIN–TERB2–TERB1$_{TRFB}$ complexes (Fig. 7a, b and Supplementary Fig. 12a, b). SEC-MALS revealed the formation of a 2:2:2:4 complex (Supplementary Fig. 12c), suggesting the recruitment of two TRF1$_{TRFH}$ dimers and consistent with the 2:1 TRF1$_{TRFH}$–TERB1$_{TRFB}$ stoichiometry.

The MAJIN$_{Core}$–TERB2–TERB1$_{TRFB}$ complex retains the DNA-binding ability of MAJIN$_{Core}$–TERB2$_C$ (Fig. 7c, d), with an apparent $K_D$ of 0.46 μM, and forms similar beads-on-a-string and plaque-like assemblies by EM (Fig. 7e). Remarkably, DNA-binding by MAJIN$_{Core}$–TERB2–TERB1$_{TRFB}$ is inhibited by TRF1$_{TRFH}$ in a manner dependent on the presence of the TRF1-binding region of TERB1 (Fig. 7f–h and Supplementary Fig. 12d), suggesting competitive inhibition of DNA-binding upon recruitment of TRF1$_{TRFH}$ by TERB1. Inclusion of MAJIN linker BP1 partly compensated for this inhibition (Fig. 7h, i and Supplementary

Fig. 12d), reminiscent of its ability to compensate for deletion of TERB2 C-terminal basic patches in DNA-binding. Accordingly, we identified a direct interaction between TERB2 C-termini (amino acids 195–220) and TRF1$_{TRFH}$ that is retained within the MAJIN$_{Core}$–TERB2$_C$ complex (Supplementary Fig. 12e), suggesting that upon recruitment by TERB1, TRF1$_{TRFH}$ binds and inhibits TERB2 C-termini, diminishing and altering DNA-binding by the wider meiotic telomere complex. Importantly, MAJIN$_{Core}$–TERB2–TERB1$_{TRFB}$ can be recruited to DNA by full length TRF1 through direct DNA-binding of its C-terminal MYB domains (Fig. 7j). Furthermore, MAJIN$_{\Delta TM}$–TERB2–TERB1$_{TRFB}$ retains its strong interaction with TRF1$_{TRFH}$ even in the presence of DNA (Fig. 7k and Supplementary Fig. 12f). We suggest that MAJIN–TERB2–TERB1–TRF1 represents a pre-displacement attachment complex in which the meiotic telomere complex interacts with telomere ends indirectly through TRF1, with direct telomere-binding partially inhibited by the presence of TRF1. This likely represents the structure present in vivo during zygotene, following TRF1-mediated attachment and prior to TRF1 displacement, that can be artificially stabilised by CDK inhibition[42].

**TRF1 displacement following telomere attachment.** How is TRF1 displaced from the meiotic telomere complex following telomere attachment? TERB1 residue T648 undergoes phosphorylation during meiotic prophase I in vivo and has been proposed to participate in TRF1 displacement by inhibiting its interaction with TERB1[28,42,44]. Whilst, we confirm that phosphomimetic mutation T648E disrupts TRF1$_{TRFH}$-binding of TERB1$_{TBM}$, we find that it is insufficient to disrupt complex formation when part of the wider TERB1$_{TRFB}$ construct (Supplementary Figs. 10b–e and 13a–d). Furthermore, a MAJIN$_{Core+BP1}$–TERB2–TERB1$_{TRFB}$ complex harbouring the TERB1 T648E mutation was able to recruit TRF1$_{TRFH}$ similarly to wild type (Fig. 7b). To explain this, we estimated relative binding affinities of TRF1$_{TRFH}$–TERB1 complexes through analysing dissociation upon serial dilution by SEC-MALS (Supplementary Fig. 13a–d). This revealed that TRF1$_{TRFH}$-binding by TERB1$_{TBM}$ is at least 100-fold weaker than TERB1$_{TRFB}$, and whilst the T648E mutation does weaken binding of TERB1$_{TRFB}$, its affinity remains at least 10-fold higher than wild-type TERB1$_{TBM}$. Further, we tested the ability of CDK1-CyclinB phosphorylation to dissociate TRF1$_{TRFH}$–TERB1 complexes (Supplementary Fig. 13e–h). TERB1$_{TBM}$ was readily phosphorylated by CDK1-CyclinB, inducing dissociation of its complex with TRF1$_{TRFH}$ (Supplementary Fig. 13e, g, h). In contrast, TERB1$_{TRFB}$ was only partially phosphorylated, suggesting steric hindrance of enzyme access, with retention of the 2:1 complex (Supplementary Fig. 13f–h). Thus, isolated phosphorylation of T648 is unlikely to be sufficient

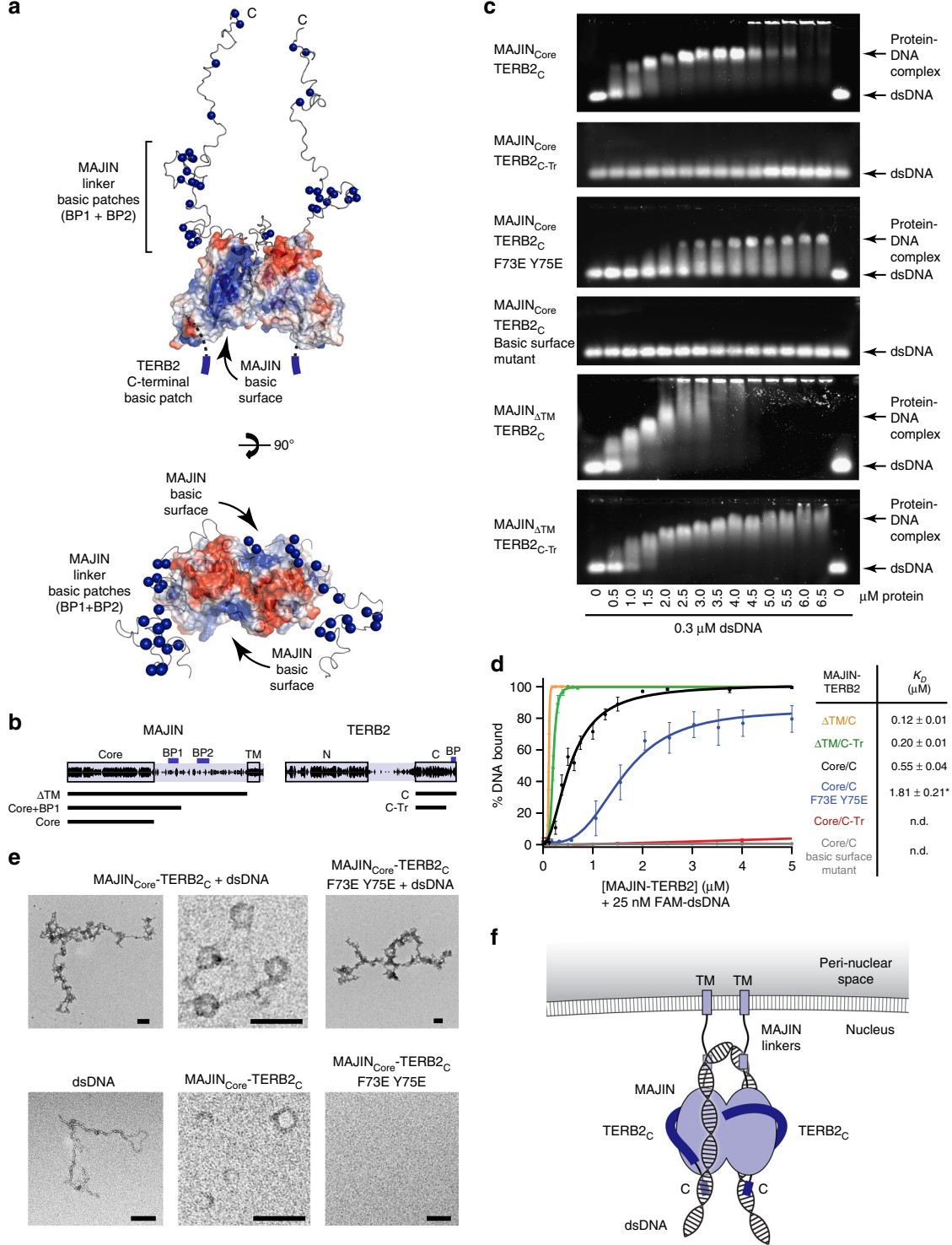

to displace TRF1 from the meiotic telomere attachment complex. Instead, additional cellular events may first loosen the TRF1–TERB1 complex to provide access for CDK1-CyclinB, and then collaborate with T648 phosphorylation to achieve TRF1 displacement.

We predict that upon TRF1 displacement, inhibition of MAJIN–TERB2–TERB1 DNA-binding is removed, enabling direct telomere-binding by the meiotic telomere complex. To test this, we visualised attachment complexes in chromosomes spread from mouse zygotene and pachytene-spermatocytes and structurally preserved pachytene nuclei (Fig. 8a–e and

Supplementary Figs. 14, 15). In zygotene, TRF1 and TERB1 show partially overlapping distributions at the telomeric ends of chromosomes (Fig. 8a and Supplementary Fig. 14a). In late pachytene, they show more distinct staining patterns, with TERB1 showing clear foci at each telomere end that are surrounded by TRF1 in a grasp-like distribution orientated laterally and proximal to the chromosome axis (Fig. 8b, c and Supplementary Figs. 14c, e, f and 15d). This is consistent with the reported TRF1 displacement following attachment[42], but rather than a complete exchange, it instead suggests a subtle remodelling in which TRF1 remains in close proximity of the meiotic telomere

**Fig. 4** DNA-binding by MAJIN–TERB2. **a** Surface electrostatic potential of the MAJIN$_{Core}$–TERB2$_C$ crystal structure (red, electronegative; blue, electropositive), with SAXS-modelled MAJIN C-terminal extensions in which basic residues are highlighted as blue spheres, and indicating the likely position of basic TERB2 C-terminal ends. Each MAJIN protomer displays a basic surface, which is continuous with the TERB2 C-terminus and basic patches towards the N-terminal end of the MAJIN flexible linkers, providing two extended basic interfaces per MAJIN–TERB2 heterotetramer. **b** Schematic of the human MAJIN and TERB2 sequences, highlighting the core linker basic patches (BP1 and BP2) and transmembrane helix of MAJIN, and N- and C-terminal domains and C-terminal basic patch (BP) or TERB2. Principal protein constructs are indicated: MAJIN ΔTM (1–233), Core + BP1 (1–147) and Core (1–112); TERB2 C (168–220) and C-Tr (168–207). **c** EMSA analysing the ability of MAJIN–TERB2 constructs (as indicated) to interact with 0.3 μM (per molecule) linear double-stranded DNA (dsDNA). Gel images are representative of at least three replicate EMSAs. **d** Quantification of DNA-binding by MAJIN–TERB2 constructs (ΔTM/C, yellow; ΔTM/C-Tr, green; Core/C, black; Core/C F73E Y75E, blue; Core/C-Tr, red; Core/C basic surface mutant, grey) through densitometry of EMSAs performed using 25 nM (per molecule) FAM-dsDNA. Plots and apparent $K_D$ values were determined by fitting data to the Hill equation; error bars indicate standard error, $n = 3$ EMSAs. *Apparent $K_D$ was estimated graphically from the concentration at 50% DNA-binding as binding saturation was not achieved. Source data are provided as a Source Data file. **e** Electron microscopy analysis of MAJIN$_{Core}$-TERB2$_C$ and MAJIN$_{Core}$–TERB2$_C$ F73E Y75E alone and in complex with plasmid dsDNA. Scale bars, 100 nm. The bottom-right panel shows lack of assembly, through absence of structures of sufficient size for EM visualisation, of MAJIN$_{Core}$–TERB2$_C$ F73E Y75E in comparison with wild-type. **f** Model of the MAJIN–TERB2 complex with dsDNA. A seamless DNA-binding interface is formed on either side of the MAJIN–TERB2 heterotetramer by the MAJIN surface, TERB2 C-terminus and MAJIN linker. A single dsDNA molecule may bind to both surfaces cooperatively through looping around the top of the molecule within the MAJIN linker region. Source data

complex and chromosome axis. In accordance with our model, MAJIN shows further separation from TRF1, localised distal with respect to the chromosome axis in spread zygotene and pachytene chromosomes and in structurally preserved pachytene nuclei (Fig. 8a, b, d and Supplementary Figs. 14b–d and 15a–c, e). Importantly, fluorescence in situ hybridisation (FISH) demonstrated the localisation of telomeric repeat DNA distal to TERB2 relative to the chromosome axis during pachytene (Fig. 8e and Supplementary Fig. 15f), consistent with it binding to the MAJIN–TERB2 core and forming extended looping structures around MAJIN linkers within the nuclear lamina.

On the basis of our combined biochemical and imaging studies, we propose that meiotic telomere attachment is achieved through a coordinated hierarchy of events that can be characterised by two molecularly distinct complexes. Firstly, TRF1 becomes recruited to the meiotic telomere complex and mediates its interaction with telomere ends (Fig. 8f). Subsequently, TRF1 is displaced, removing inhibition of MAJIN–TERB2–TERB1 DNA-binding and enabling formation of a post-displacement complex, in which telomere ends are bound directly by the meiotic telomere complex (Fig. 8g).

## Discussion

The MAJIN–TERB2–TERB1 meiotic telomere complex integrates functions of the LINC and shelterin complexes to physically connect telomere ends to the cytoskeleton. Here, we report the crystal structure of MAJIN–TERB2, constituting the architectural core of the meiotic telomere complex. This 2:2 heterotetramer is formed of a globular dimer of MAJIN β-grasp domains encircled by two TERB2 chains. β-grasp folds were first identified in ubiquitin and display an unusually high degree of structural and functional diversity[47,48]. The topology adopted by MAJIN is to our knowledge unique, consisting of a five-stranded assemblage with a two-stranded β-sheet insertion. TERB2 wraps around the β-grasp fold, completing the closure of the central α-helix and providing additional β-strands to the grasping and appending β-sheets. The 2:2 complex is formed through aromatic interactions in a MAJIN dimerization interface; similar dimer formation has been observed in other β-grasp proteins, albeit through diverse means such as domain swap or metal coordination[48]. MAJIN is highly unstable in the absence of TERB2, indicating that its presence is necessary for folding of the β-grasp domain, and thus the MAJIN–TERB2 complex is likely constitutive in vivo.

MAJIN–TERB2 provides extensive DNA-binding interfaces that extend from TERB2 C-termini, across MAJIN β-sheet surfaces to MAJIN linkers, and demonstrate cooperativity between interfaces on either side of the molecule. On this basis, we

propose that individual telomeric DNA chains loop around the top of MAJIN–TERB2 molecules to enable their interaction with both interfaces. The MAJIN linkers constitute flexible extensions from the structural core to C-terminal transmembrane helices, which under tension may stretch up to 400 Å to provide a separation between the core and the inner nuclear membrane that spans the nuclear lamina. Whilst a previous study reported that MAJIN linker basic patches are essential for DNA-binding[42], we find that their inclusion provided an enhancement in DNA-binding affinity from an apparent $K_D$ of 0.55 μM for MAJIN$_{Core}$–TERB2$_C$ to 0.12 μM for MAJIN$_{ΔTM}$–TERB2$_C$. A simple explanation is that the previous study analysed isolated MAJIN, whereas TERB2 is essential for MAJIN folding, so may have detected residual binding of basic residues in absence of a folded core. Nevertheless, their mutation fails to fully rescue telomere attachment defects of $Majin^{-/-}$ spermatocytes[42]. Thus, we suggest that MAJIN linker basic patches may contact and stabilise telomere loops between MAJIN–TERB2 attachments, facilitating their extension into the nuclear lamina towards the inner nuclear membrane. This raises the intriguing possibility that a three-dimensional assembly incorporating MAJIN linkers and telomere loops contributes to the mechanism, whereby the meiosis-specific nuclear lamina enables the rapid and fluid movement of telomere ends through the nuclear envelope[27].

The wider meiotic telomere complex forms a 2:2:2 assembly of MAJIN–TERB2–TERB1, in which the 2:2 MAJIN–TERB2$_C$ structure core is connected to two globular 1:1 TERB2$_N$–TERB1$_{TRFB}$ complexes through long-flexible linkers within TERB2. These TERB2 linkers can stretch to ~200 Å under tension, accounting for the physical separation observed in mouse pachytene spermatocyte chromosomes. Each TERB2$_N$–TERB1$_{TRFB}$ complex can recruit one TRF1 dimer, leading to the formation of a 2:2:2:4 meiotic telomere recruitment complex of MAJIN–TERB2–TERB1–TRF1. Whilst crystal structures have shown that a TRF1 dimer is capable of binding to two interacting peptides from TERB1 (TBM; amino acids 642–656)[44,45], we find that a wider region of TERB1 (TRFB; amino acids 568–658) is required for stable complex formation in solution. This likely blocks the second site through steric hindrance to provide a 2:1:1 TRF1–TERB1–TERB2 stoichiometry, analogous to the 2:1:1:1 stoichiometry observed for the TRF1–TIN2–TPP1–POT1 shelterin complex[41]. This is supported by our observation that TERB1$_{TRFB}$ exhibits a greater than 100-fold increased binding affinity for TRF1$_{TRFH}$ than TERB1$_{TBM}$, and the previous finding that the TERB1$_{TRFB}$ interaction is mediated by a single TRF1$_{TRFH}$–TERB1$_{TBM}$-binding interface[45]. Further, previous studies reported $K_D$ values for TRF1$_{TRFH}$-binding of 5.6 μM and 75 nM for free TERB1$_{TBM}$ and GST–TERB1$_{TBM}$, respectively[44,45]. The higher apparent affinity in the latter case is likely due to the

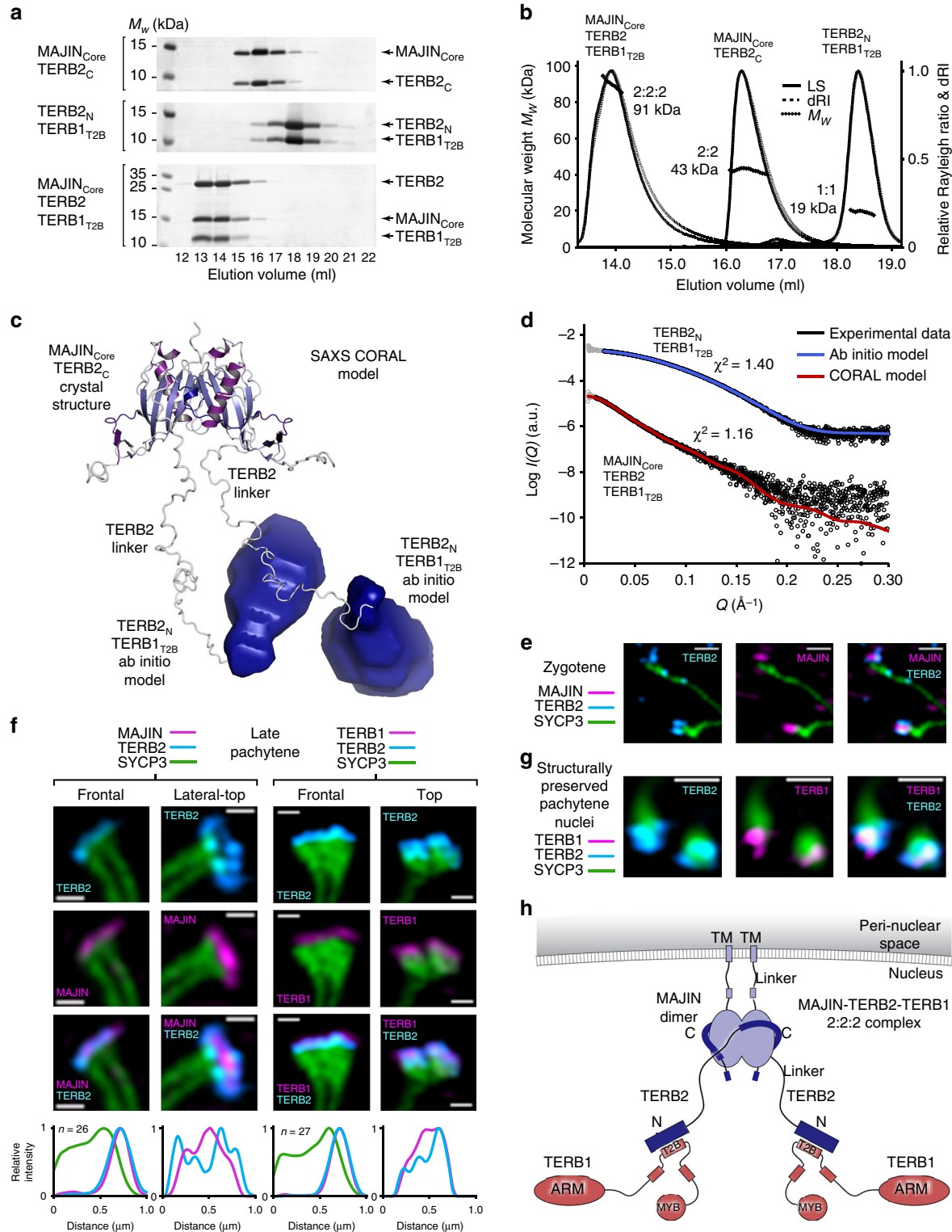

cooperativity resulting from simultaneous binding of two peptides of a GST-induced dimer at both sites within a TRF1$_{TRFH}$ dimer. We propose that a similar two-site cooperative binding mode may be exhibited by TERB1$_{TRFB}$, with an N-terminal sequence (amino acids 561–585) interacting with a second binding site within the TRF1$_{TRFH}$ dimer in a manner that may partially mimic TERB1$_{TBM}$, thereby restricting the complex to a 2:1 stoichiometry. The recruitment of two TRF1 dimers by each MAJIN–TERB2–TERB1 complex likely provides the means of telomere recruitment,

bringing telomeric DNA into close proximity that enables its loading onto MAJIN–TERB2–TERB1 complexes.

TRF1 is displaced proximally towards the chromosome axis following telomere attachment. CDK phosphorylation site T648 has been identified within the TRF1-binding site of TERB1[42], and T648E phosphomimetic mutation blocks the interaction of short peptide TERB1$_{TBM}$ with TRF1[44]. However, we find that the same mutation within the context of the full TRF1-interacting region TERB1$_{TRFB}$ is insufficient to disrupt the complex and simply

**Fig. 5** Structure of the 2:2:2 MAJIN–TERB2–TERB1 meiotic telomere complex. **a** Size-exclusion chromatography elution profiles of MAJIN$_{Core}$-TERB2$_C$ (1–112, 168–220), TERB2$_N$-TERB1$_{T2B}$ (1–107, 585–642) and MAJIN$_{Core}$-TERB2–TERB1$_{T2B}$ (1–112, 1–220, 585–642). **b** SEC-MALS analysis. MAJIN$_{Core}$-TERB2$_C$ is 2:2 (43 kDa; theoretical 2:2 – 40 kDa), TERB2$_N$-TERB1$_{T2B}$ is 1:1 (19 kDa; theoretical 1:1–20 kDa) and MAJIN$_{Core}$-TERB2–TERB1$_{T2B}$ is 2:2:2 (91 kDa; theoretical 2:2:2–94 kDa). **c, d** SEC-SAXS analysis of MAJIN$_{Core}$-TERB2–TERB1$_{T2B}$ and TERB2$_N$-TERB1$_{T2B}$. **c** SAXS CORAL model of MAJIN$_{Core}$-TERB2–TERB1$_{T2B}$ constructed by rigid body fitting of the MAJIN$_{Core}$-TERB2$_C$ crystal structure and TERB2$_N$-TERB1$_{T2B}$ ab initio models (generated from TERB2$_N$-TERB1$_{T2B}$ SAXS data; Supplementary Fig. 7b,d,e), with modelling of the linking TERB2 sequence, to experimental SAXS data. **d** SAXS scattering data for TERB2$_N$-TERB1$_{T2B}$ and MAJIN$_{Core}$-TERB2–TERB1$_{T2B}$ overlaid with theoretical scattering curves of the ab initio model (red; $\chi^2 = 1.40$) and CORAL model (grey; $\chi^2 = 1.16$), respectively. The scattering data points below the minimum $Q$-value of the Guinier analysis are shown in grey. **e–g** Structured illumination microscopy. Wide field images, plot analyses and additional orientations are shown in Supplementary Figs. 8 and 9. **e** Spread mouse zygotene spermatocyte chromosomes stained with anti-SYCP3 (green), anti-TERB2 (cyan) and anti-MAJIN (magenta). Scale bars, 0.3 µm. **f** Spread mouse pachytene spermatocyte chromosomes stained with anti-SYCP3 (green), anti-TERB2 (cyan) and anti-MAJIN (left) or anti-TERB1 (right) (magenta). Scale bars, 0.3 µm. Normalised intensity-distance plots are shown; frontal plots represent averages of multiple images (n = 26, n = 27 telomeres), other orientation plots represent individual images. Source data are provided as a Source Data file. **g** Structurally preserved mouse spermatocyte pachytene nuclei stained with anti-SYCP3 (green), anti-TERB2 (cyan) and anti-TERB1 (magenta). Scale bars, 0.5 µm. **h** Model of the 2:2:2 MAJIN–TERB2–TERB1 complex. The central MAJIN–TERB2 2:2 core is tethered to the nuclear membrane through long-flexible MAJIN linkers leading to C-terminal transmembrane helices that are embedded in the inner nuclear membrane. Distally, it is tethered to two distinct TERB2$_N$-TERB1 complexes through long-flexible linkers within TERB2.Source data

weakens TRF1$_{TRFH}$-binding to an affinity that is more than 10-fold stronger than wild-type TERB1$_{TBM}$. Thus, T648 phosphorylation must be combinatorial with other phosphorylation/binding events within TERB1 and/or surrounding proteins, coupled with other signalling mechanisms, to achieve TRF1 displacement, in keeping with the failure of a T648A mutation to prevent TRF1 displacement in vivo[42].

The generation of rapid prophase movements of over 100 nm s$^{-1}$ necessitates substantial force transmission along the meiotic telomere-LINC axis[6]. This is achieved across the nuclear membrane through a parallel organisation of LINC complexes that contact the same microtubule via dynein–dynactin, each transmitting small forces that summate to generate coordinated unidirectional movements[6]. The parallel forces transmitted along LINC complexes must converge, but does this occur upon telomere attachment or through prior integration within the meiotic telomere complex? The former model involves individual interactions of meiotic telomere complexes with LINC complexes and telomeric DNA that are summated within chromosomes. The latter model involves a higher order assembly of meiotic telomere complexes that receives interactions from multiple LINC complexes and forms a single telomere attachment. Notably, the MAJIN$_{Core}$–TERB2$_C$ P3$_2$21 crystal lattice includes contacts between TERB2 C-termini that form continuous linear chains of MAJIN–TERB2 complexes (Supplementary Fig. 16a). Whilst not stable in solution, we wonder whether such contacts may generate large molecular assemblies upon concentration on the inner nuclear membrane in vivo (Supplementary Fig. 16b). In these structures, telomeric DNA may undergo extensive looping around long chains of MAJIN–TERB2 molecules to achieve single robust attachments that span the width of the axis and integrate forces transmitted from attached LINC complexes (Supplementary Fig. 16c). This model may explain the circular assemblies of MAJIN$_{Core}$–TERB2$_C$ observed by electron microscopy, which became thickened and linked together in a beads-on-a-string fashion upon binding to DNA.

What is the role of shelterin following telomere attachment? A previous study proposed a cap exchange model in which shelterin is released from telomeres and remains in close proximity to enable rebinding upon detachment[42]. However, an alternative model is that shelterin is never released but merely displaced to surrounding unattached telomeric DNA. The length of telomeric DNA (10–15 kb in humans[35]) is sufficient to account for this model and is consistent with the localisation of TRF1 in close proximity to the chromosome axis by SIM. Indeed, displacement may represent a remodelling event, in which TRF1/shelterin and assembled MAJIN–TERB2–TERB1 form interlaced interactions with telomeric DNA to create cooperative and robust telomere end attachments.

Upon synapsis, telomere attachments mature into ornate end attachment plates, in which the synaptonemal complex seamlessly fuses with the nuclear envelope[15–18], raising the question of how meiotic telomere attachment is integrated with chromosome structure? A crucial link is found in the C-terminal MYB domain of TERB1, which recruits cohesins to telomere ends to impose their structural rigidity[28]. Similarly, synaptonemal complex protein SYCP3 is required to form the iconic conical axis thickenings within attachment plates[17]. Thus, whilst not participating in telomere attachment per se, meiotic cohesins and axis proteins are required to impose a chromosomal structure that facilitates nuclear envelope attachment[17,28,49,50]. We envisage that meiotic telomere complex assembly with telomeric DNA may integrate with meiotic cohesins, axis proteins, and potentially the SC midline lattice, in addition to LINC and shelterin complexes, to form an ordered supramolecular complex that achieves the coordinated attachment of synapsed telomere ends to the nuclear envelope.

## Methods

**Recombinant protein expression and purification.** Sequences corresponding to regions of human *TERB1*, *TERB2*, *MAJIN* and *TRF1* were cloned into pHAT4[51], pMAT11[51] or pRSF-Duet1 (Merck Millipore) vectors for expression as TEV-cleavable His$_6$-, His$_6$-MBP or MBP-fusion proteins, respectively. A list of primers used for cloning is provided in Supplementary Table 2. Constructs were co-expressed in BL21 (DE3) cells (Novagen®), in 2xYT media, induced with 0.5 mM IPTG for 16 h at 15 °C. Cell disruption was achieved by sonication in 20 mM Tris pH 8.0, 500 mM KCl and cellular debris removed by centrifugation at 40,000 *g*. Fusion proteins were purified through consecutive Ni-NTA (Qiagen), amylose (NEB) and HiTrap Q HP (GE Healthcare) ion exchange chromatography. Ni-NTA chromatography was performed in lysis buffer, with elution in 20 mM Tris pH 8.0, 500 mM KCl, 200 mM imidazole. Amylose affinity chromatography was performed in 20 mM Tris pH 8.0, 150 mM KCl, 2 mM DTT, with elution through addition of 30 mM D-maltose. Ion exchange chromatography was performed through a linear salt gradient of 20 mM Tris pH 8.0, 100–1000 mM KCl, 2 mM DTT. TEV protease was utilised to remove affinity tags, and cleaved samples were purified through ion exchange chromatography in 20 mM Tris pH 8.0, 100–1000 mM KCl, 2 mM DTT, and size exclusion chromatography (HiLoad™ 16/600 Superdex 200, GE Healthcare) in 20 mM Tris pH 8.0, 250 mM KCl, 2 mM DTT. Protein samples were concentrated using Microsep™ Advance Centrifugal Devices 10,000 MWCO centrifugal filter units (PALL) and were stored at −80 °C following flash-freezing in liquid nitrogen. Purification of TRF1 and TRF1-containing complexes utilised the lysis, Ni-NTA, amylose, ion exchange and gel filtration buffers described above, with the addition of 10% glycerol, and was performed at room temperature to prevent protein precipitation. Protein samples were analysed by SDS–PAGE and visualised with Coomassie staining. Concentrations were determined by UV spectroscopy using a Cary 60 UV spectrophotometer (Agilent) with extinction coefficients and molecular weights calculated by ProtParam (http://web.expasy.org/protparam/). The MAJIN basic surface mutant K24M K26E R28E K31D R34E R81D was designed using the ROSIE Rosetta Sequence Tolerance Server[52].

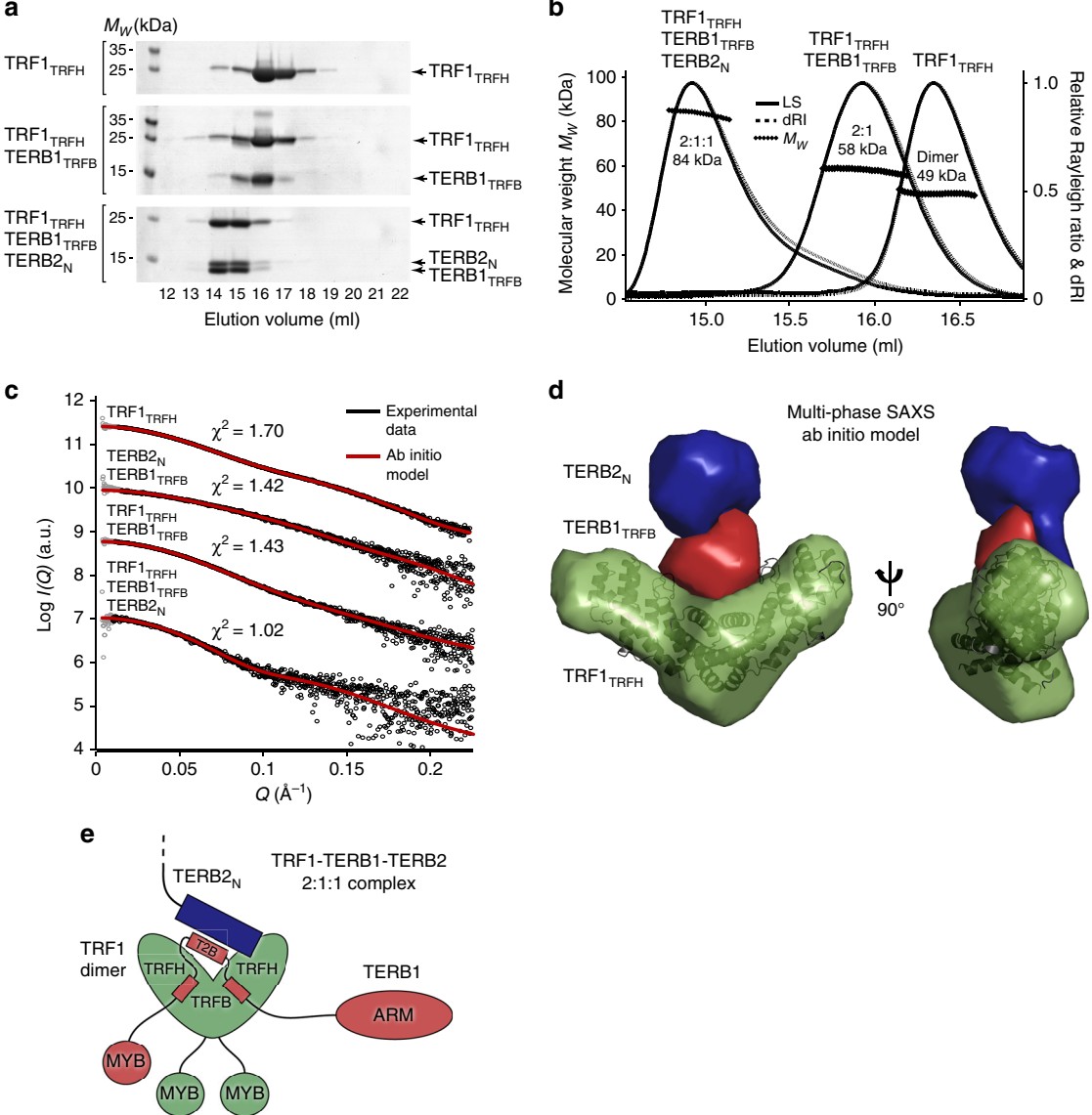

**Fig. 6** Structure of the 2:1:1 TRF1–TERB1–TERB2 complex. **a** Size-exclusion chromatography elution profiles of TRF1$_{TRFH}$ (62–268), TRF1$_{TRFH}$-TERB1$_{TRFB}$ (62–268, 561–658) and TRF1$_{TRFH}$-TERB1$_{TRFB}$-TERB2$_N$ (62–268, 561–658, 1-119). A longer TERB2$_N$ construct of 1-119 was used for size exclusion chromatography and SDS–PAGE to provide molecular weight discrimination from TERB1$_{TRFB}$; all other experiments use a shorter 1–107 construct. Source data are provided as a Source Data file. **b** SEC-MALS analysis. TRF1$_{TRFH}$ is a dimer (49 kDa; theoretical dimer – 48 kDa), TRF1$_{TRFH}$-TERB1$_{TRFB}$ is 2:1 (58 kDa; theoretical 2:1–59 kDa) and TRF1$_{TRFH}$-TERB1$_{TRFB}$-TERB2$_N$ is 2:1:1 (84 kDa; theoretical 2:1:1–73 kDa). **c, d** Multi-phase SAXS ab initio modelling of TRF1$_{TRFH}$-TERB1$_{TRFB}$-TERB2$_N$ using MONSA in which experimental data of the complex and its constituents were used to fit their ab initio structures and relative orientations within the full complex. **c** SAXS scattering data of TRF1$_{TRFH}$, TERB2$_N$-TERB1$_{TRFB}$, TRF1$_{TRFH}$-TERB1$_{TRFB}$ and TRF1$_{TRFH}$-TERB1$_{TRFB}$-TERB2$_N$ overlaid with the theoretical scattering curves of their modelled structures (red; $\chi^2$ values of 1.70, 1.42, 1.43 and 1.02, respectively). The scattering data points below the minimum Q-value of the Guinier analysis are shown in grey. **d** Multi-phase SAXS model of TRF1$_{TRFH}$-TERB1$_{TRFB}$-TERB2$_N$ in which TRF1$_{TRFH}$ is shown in green with its known crystal structure superposed (PDB accession 5WIR[45]), and TERB1$_{TRFB}$ and TERB2$_N$ are shown in red and blue, respectively. **e** Model of the 2:1:1 TRF1–TERB1–TERB2 complex. The dimeric cleft of the TRF1 TRFH domains interacts with TERB1 through the TRFB region that includes its binding site for the globular N-terminus of TERB2. Source data

**Recombinant expression and purification of full length TRF1**. TRF1 (1–439) was expressed as described above, and cell lysis was performed in 20 mM Tris pH 8.0, 500 mM KCl, 10% glycerol. Ni-NTA (Qiagen) affinity and HiTrap Q HP (GE Healthcare) ion exchange chromatography were performed using the same buffer, the latter with a salt gradient between 0.1–1.0 M KCl. TEV protease was added at a 1:15 ratio and incubated overnight at room temperature. Cleaved TRF1 was further purified through HiTrap Q HP ion exchange chromatography (salt gradient 0.1–1.0 M KCl) and size exclusion chromatography (HiLoad™ 16/600 Superdex 200, GE Healthcare) in 20 mM Tris pH 8.0, 250 mM KCl, 10% glycerol buffer, 2 mM DTT. TRF1 was concentrated to a final concentration of 15 mg ml$^{-1}$ using Microsep™ Advance Centrifugal Devices 10,000 MWCO centrifugal filter units (PALL) at room temperature and stored at −80 °C following flash-freezing in liquid nitrogen.

**Preparation of a MAJIN−TERB2 selenomethionine derivative**. Transformed BL21 (DE3) *E. coli* were cultured in 2xYT media (Formedium), harvested at an OD$_{600}$ of 0.6 and washed in 150 mM NaCl before being transferred to M9 media (Formedium) supplemented with trace elements (2.5 mg l$^{-1}$ CoCl$_2$.6H$_2$O, 15 mg l$^{-1}$ MnCl$_2$.4H$_2$O, 1.5 mg l$^{-1}$ CuCl$_2$.2H$_2$O, 3 mg l$^{-1}$ H$_3$BO$_3$, 33.8 mg l$^{-1}$ Zn (CH$_3$COO)$_2$.2H$_2$O and 14.10 mg l$^{-1}$ TitriplexIII) and 5 μM Zn(OAc)$_2$. Methionine biosynthesis was inhibited by addition of 100 mg l$^{-1}$ lysine, 100 mg l$^{-1}$ phenylalanine, 100 mg l$^{-1}$ threonine, 50 mg l$^{-1}$ isoleucine, 50 mg l$^{-1}$ leucine, 50 mg l$^{-1}$ valine. After 1 h at 25 °C, 250 rpm, the cultures were supplemented with 50 mg l$^{-1}$ seleno-methionine. Expression was induced after 30 min by 0.5 mM IPTG and incubated overnight at 15 °C, 250 rpm. Selenomethionine MAJIN$_{Core}$−TERB2$_C$ was purified as described for the native protein.

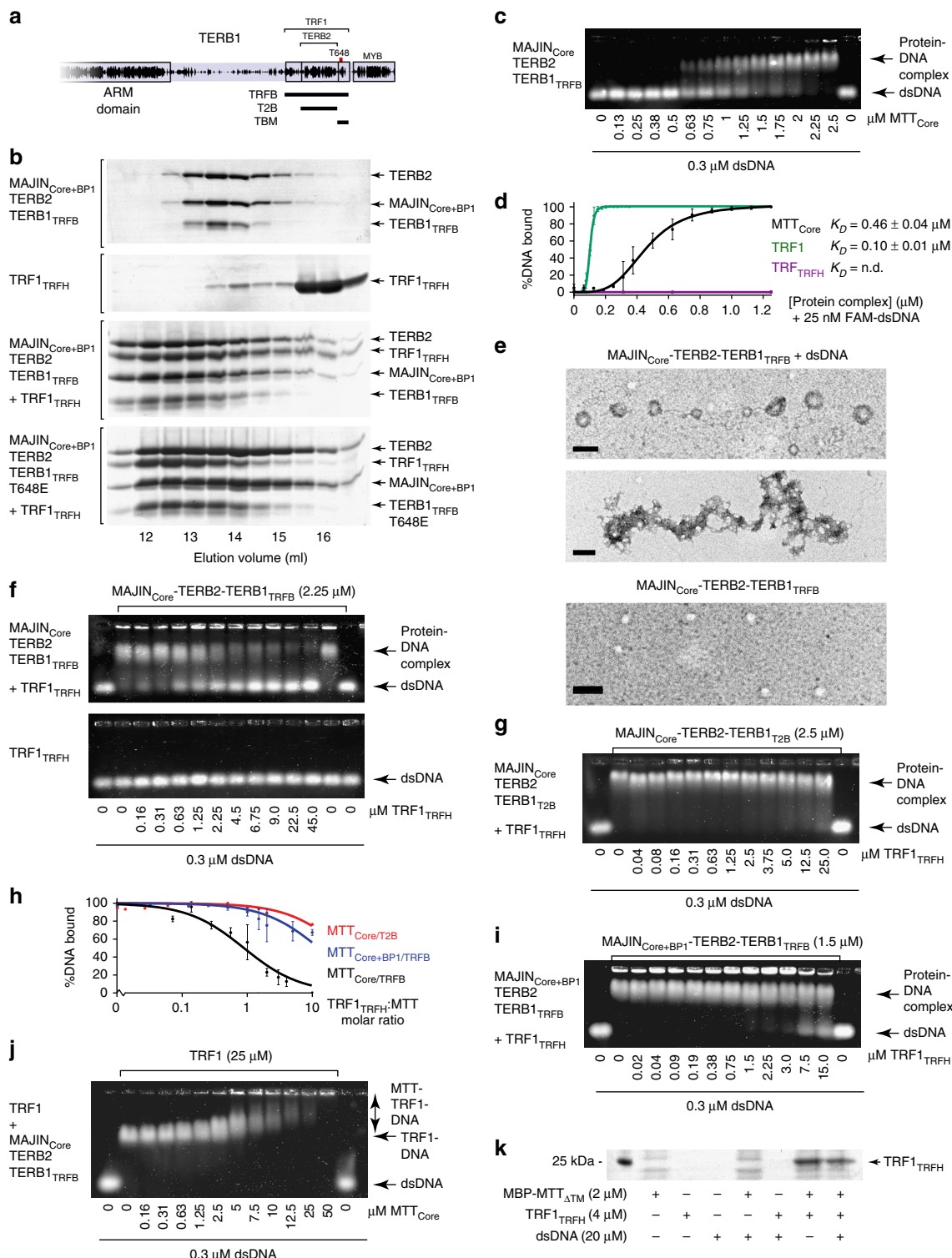

**Circular dichroism spectroscopy**. Far UV circular dichroism (CD) spectroscopy data were collected on a Jasco J-810 spectropolarimeter (Institute for Cell and Molecular Biosciences, Newcastle University). Buffer subtracted CD spectra were recorded using a 0.2 mm pathlength cuvette (Hellma) in 10 mM $Na_2HPO_4$/ $NaH_2PO_4$ pH 7.5, 250 mM NaF between 260 and 185 nm. Data were collected at 20 nm min$^{-1}$ with a pitch of 0.2 nm, smoothed with a response time of 4 s and plotted as mean residue ellipticity ([θ]) (x1000 deg.cm$^2$.dmol$^{-1}$.residue$^{-1}$). The CDSSTR algorithm of the Dichroweb server (http://dichroweb.cryst.bbk.ac.uk) was utilised to estimate secondary structure content of analysed samples. Protein stability was assessed through thermal denaturation in 20 mM Tris pH 8.0, 250 mM KCl, 2 mM DTT using a 1 mm pathlength quartz cuvette (Hellma). Data were recorded at 222 nm, between 5 °C and 95 °C at 2 °C per min every 0.5 °C, and were

converted to mean residue ellipticity ([θ]$_{222}$) and plotted as % unfolded ([θ]$_{222,x}$– [θ]$_{222,5}$)/([θ]$_{222,95}$–[θ]$_{222,5}$). Melting temperatures (Tm) were estimated as the points at which samples are 50% unfolded.

**Size-exclusion chromatography multi-angle light scattering**. The absolute molar masses of protein samples and complexes were determined by size-exclusion chromatography multi-angle light scattering (SEC-MALS). Protein samples at > 1 mg ml$^{-1}$ were loaded onto a Superdex™ 200 Increase 10/300 GL size exclusion chromatography column (GE Healthcare) in 20 mM Tris pH 8.0, 250 mM KCl, 2 mM DTT, at 0.5 ml min$^{-1}$ using an ÄKTA™ Pure (GE Healthcare). The column outlet was fed into a DAWN® HELEOS™ II MALS detector (Wyatt Technology),

**Fig. 7** TRF1 recruitment by MAJIN–TERB2–TERB1. **a** Schematic of the human TERB1 sequence, highlighting the TRF1/TERB2-interacting region and phosphorylation site T648, aligned with constructs TRFB (561–658), T2B (585–642) and TBM (642–658). **b** Size-exclusion chromatography elution profiles of MAJIN$_{Core+BP1}$-TERB2–TERB1$_{TRFB}$ (1–147, 1–220, 561–658), TRF1$_{TRFH}$ (62–268), and MAJIN$_{Core+BP1}$-TERB2–TERB1$_{TRFB}$ WT and T648E (TERB1) upon incubation with a stoichiometric amount of TRF1$_{TRFH}$. **c** EMSA analysing the ability of MAJIN$_{Core}$-TERB2–TERB1$_{TRFB}$ to interact with linear double-stranded DNA (dsDNA). Gel images are representative of at least three replicate EMSAs. **d** Quantification of DNA-binding by MAJIN$_{Core}$-TERB2–TERB1$_{TRFB}$ (MTT$_{Core}$, black), TRF1 (green) and TRF1$_{TRFH}$ (purple) through densitometry of EMSAs performed using 25 nM (per molecule) FAM-dsDNA. Plots and apparent $K_D$ values were determined by fitting data to the Hill equation; error bars indicate standard error, $n = 3$ EMSAs. Source data are provided as a Source Data file. **e** Electron microscopy analysis of MAJIN$_{Core}$-TERB2–TERB1$_{TRFB}$ alone and in complex with plasmid dsDNA. Scale bars, 100 nm. **f** EMSA of MAJIN$_{Core}$-TERB2–TERB1$_{TRFB}$ upon incubation with TRF1$_{TRFH}$ (top) and TRF1$_{TRFH}$ alone (bottom) with linear double-stranded DNA (dsDNA). **g** EMSA of MAJIN$_{Core}$-TERB2–TERB1$_{T2B}$ with linear dsDNA upon incubation with TRF1$_{TRFH}$. **h** Quantification of DNA-binding by MAJIN$_{Core}$-TERB2–TERB1$_{TRFB}$ (MTT$_{Core/TRFB}$, black), MAJIN$_{Core}$-TERB2–TERB1$_{T2B}$ (MTT$_{Core/T2B}$, red) and MAJIN$_{Core+BP1}$-TERB2–TERB1$_{TRFB}$ (MTT$_{Core+BP1/TRFB}$, blue) upon incubation with TRF1$_{TRFH}$ (at molar ratios indicated) through densitometry of EMSAs; error bars indicate standard error, $n = 3$ EMSAs. **i** EMSA of MAJIN$_{Core+BP1}$-TERB2–TERB1$_{TRFB}$ with linear dsDNA upon incubation with TRF1$_{TRFH}$. **j** EMSA of TRF1 (full length) with linear dsDNA upon incubation with MAJIN$_{Core}$-TERB2–TERB1$_{TRFB}$. **f–j** Gel images and plots are representative of at least three replicate EMSAs. **k** Amylose pulldown of TRF1$_{TRFH}$ using MBP-fusion MAJIN$_{ΔTM}$-TERB2–TERB1$_{TRFB}$ (MBP-MTT$_{ΔTM}$) with or without pre-incubation with plasmid dsDNA (as indicated). The uncropped gel image is shown in Supplementary Fig. 12f. Source data

---

followed by an Optilab® T-rEX™ differential refractometer (Wyatt Technology). Light scattering and differential refractive index data were collected and analysed using ASTRA® 6 software (Wyatt Technology). Molecular weights and estimated errors were calculated across eluted peaks by extrapolation from Zimm plots using a $dn/dc$ value of 0.1850 ml g$^{-1}$. SEC-MALS data are presented with light scattering (LS) and differential refractive index (dRI) profiles, with fitted molecular weights ($M_W$) plotted across elution peaks.

**Size-exclusion chromatography small-angle X-ray scattering.** SEC-SAXS experiments were performed at beamline B21 of the Diamond Light Source synchrotron facility (Oxfordshire, UK). Protein samples at concentrations > 10 mg/ml were loaded onto a Superdex™ 200 Increase 10/300 GL size exclusion chromatography column (GE Healthcare) in 20 mM Tris pH 8.0, 250 mM KCl at 0.5 ml min$^{-1}$ using an Agilent 1200 HPLC system. The column outlet was fed into the experimental cell, and SAXS data were recorded at 12.4 keV, detector distance 4.014 m, in 3.0 s frames. The experimental cell and beam dimensions provide an illuminated volume of 1.8 µl, corresponding to protein exposure for 0.225 s, which equates to sample exposure to ~20% of the full beam. Data were subtracted using background frames from the same SEC run either before or after the protein elution (indicated by a flat region in the plot of the integral of the ratio for each frame and following their individual inspection), averaged and analysed for Guinier region $Rg$ using ScÅtter 3.1 (http://www.bioisis.net). In all cases, Guinier regions were essentially linear, albeit with some fluctuations on either side of the linear fit dependent on the signal:noise ratio of data collection for individual constructs, indicating the presence of no more than minor levels of sample aggregation. Data below the minimum $Q$-value of the Guinier region, which are affected by proximity of the beam stop, were excluded from subsequent analysis. Approximate parameters for real space analysis were determined using the server www.bayesapp.org, and $P(r)$ distributions fitted using PRIMUS[53]. Ab initio modelling was performed using DAMMIF. Thirty independent runs were performed in P1 or P2 symmetry (as indicated) and averaged. Crystal structures were docked into DAMFILT molecular envelopes using SUPCOMB[54]. Multi-phase SAXS ab initio modelling was performed using MONSA;[55] rigid body and linker modelling was performed using CORAL[56]. Crystal structures and models were fitted to experimental data using CRYSOL[57].

**Electrophoretic mobility shift assay.** To demonstrate overt differences in DNA-binding capability between complexes, protein complexes were incubated with 0.3 µM (per molecule) 57 or 75 bp linear random sequence dsDNA, 54 bp telomeric (TTAGGG repeat) hairpin DNA (6 nt linker of CGACGA), 100 nt poly(dT) or 90 nt random sequence ssDNA substrate at concentrations indicated, in 20 mM Tris pH 8.0, 250 mM KCl for 1 h at 4 °C. For the inhibition of DNA-binding assays, 57 bp random sequence dsDNA was incubated with MAJIN–TERB2–TERB1 for 20 min at 4 °C prior to the addition of TRF1$_{TRFH}$. In DNA super-shift assays, 57 bp random sequence dsDNA was incubated with TRF1 for 20 min at 4 °C prior to the addition of MAJIN–TERB2–TERB1. Glycerol was added at a final concentration of 3%, and samples were analysed by electrophoresis on a 0.5% (w/v) agarose gel in 0.5x TBE pH 8.0 at 20 V for 4 h at 4 °C. DNA was detected by SYBR™ Gold (ThermoFisher). DNA sequences are provided in Supplementary Table 3.

**Apparent $K_D$ determination by EMSA.** Quantification of DNA-binding was performed though EMSA (as described above) using 25 nM FAM-labelled 145 bp random sequence dsDNA, 100 nt poly(dT) or 90 nt random sequence ssDNA at protein concentrations indicated (referring to the molecular oligomeric species). DNA was detected by FAM and SYBR™ Gold (ThermoFisher) staining using a Typhoon™ FLA 9500 (GE Healthcare), with 473 nm laser at excitation wavelength

490 nm and emission wavelength 520 nm, using the LPB filter and a PMT voltage of 400 V. Gels were analysed using ImageJ software (https://imagej.nih.gov/ij/). The DNA-bound proportion was plotted against molecular protein concentration and fitted to the Hill equation (below), with apparent $K_D$ determined, using Prism8 (GraphPad). Protein concentrations used for apparent $K_D$ estimation are quoted for the oligomeric species. DNA sequences are provided in Supplementary Table 3.

$$\% \, bound \, DNA = \frac{C^n}{K_D^n + C^n} \qquad (1)$$

**Electron microscopy.** Electron microscopy (EM) was performed using an FEI Philips CM100 transmission electron microscope at the Electron Microscopy Research Services, Newcastle University. Protein samples at 5–10 µM were incubated with 10 µM (per base pair) plasmid double-stranded DNA in 20 mM Tris pH 8.0, 250 mM KCl for 10 min, and applied to carbon-coated EM grids. Negative staining was performed using 2% (wt/vol) uranyl acetate.

**Amylose pulldown assay.** MAJIN$_{ΔTM}$–MBP–TERB2–TERB1$_{TRFH}$ (2 µM) was pre-incubated with plasmid dsDNA (20 µM per base pair; 7987 bp) for 30 min at 4 °C, and then with TRF1$_{TRFH}$ (4 µM) for 30 min at 4 °C in 20 mM Tris pH 8.0, 250 mM KCl, 10 mg ml$^{-1}$ BSA (100 µl reaction volume). Reactions were added to 40 µl of pre-equilibrated amylose resin (NEB) and incubated for 1 h at 4 °C. After centrifugation at 4000 $g$, the supernatant was discarded and the resin was washed twice using buffer with BSA present, and a further four times without BSA. 50 µl 1.5x LDS loading buffer (ThermoFisher) was added to resin, incubated at 95 °C for 5 min and samples were analysed by SDS–PAGE.

**TRF1$_{TRFH}$–TERB1 phosphorylation assays.** Co-purified complexes between His$_6$–TRF1$_{TRFH}$ and MBP–TERB1$_{TRFB}$ (9 µM) or MBP–TERB1$_{TBM}$ (130 µM) were incubated with 9 µM phosphorylated CDK1-CyclinB (prepared as previously described[58]) in 40 mM Tris pH 7.5, 250 mM KCl, 20 mM MgCl$_2$, 2 mM ATP pH 7.5, 10 mM DTT for 10 min at 30 °C and a further 50 min at 22 °C. Samples were analysed by SEC-MALS and SDS–PAGE with staining by Coomassie and ProQ-Diamond (ThermoFisher), for which imaging was performed using a Typhoon™ FLA 9500 (GE Healthcare), with 532 nm laser, excitation wavelength 555 nm and emission wavelength 580 nm, using the LPG filter and a PMT voltage of 400 V.

**Structure solution of MAJIN$_{1-112}$–TERB2$_{168-220}$.** MAJIN$_{1-112}$–TERB2$_{168-220}$ selenomethionine derivative protein crystals were obtained through vapour diffusion in sitting drops, by mixing 100 nl of protein at 16 mg ml$^{-1}$ with 100 nl of crystallisation solution (0.12 M 1,6-hexanediol; 0.12 M 1-butanol; 0.12 M 1,2-propanediol (racemic); 0.12 M 2-propanol; 0.12 M 1,4-butanediol; 0.12 M 1,3-propanediol; 55.5 mM MES pH 3.11, 44.5 mM imidazole pH 10.23; 12.5% w/v PEG 1000; 12.5% w/v PEG 3350; 12.5% v/v MPD) and equilibrating at 20 °C. Crystals grew overnight, were harvested after 1 week and flash frozen in liquid nitrogen. X-ray diffraction data were collected at 0.9159 Å, 100 K, as three datasets of the same crystal (at 15° kappa increments), each of 3600 consecutive 0.10° frames of 0.050 s exposure on a Pilatus 6 M detector at beamline I04-1 of the Diamond Light Source synchrotron facility (Oxfordshire, UK). Data were indexed and integrated in XDS[59], and datasets were scaled together in XSCALE[60] and merged in Aimless[61]. Crystals belong to trigonal spacegroup P3$_2$21 (cell dimensions a = 59.88 Å, b = 59.88 Å, c = 159.93 Å, α = 90°, β = 90°, γ = 120°), with a 2:2 MAJIN–TERB2 heterotetramer in the asymmetric unit. Structure solution was achieved through SAD experimental phasing utilising the anomalous signal of incorporated selenium atoms. Heavy atom sites (eight) were identified, and a density-modified

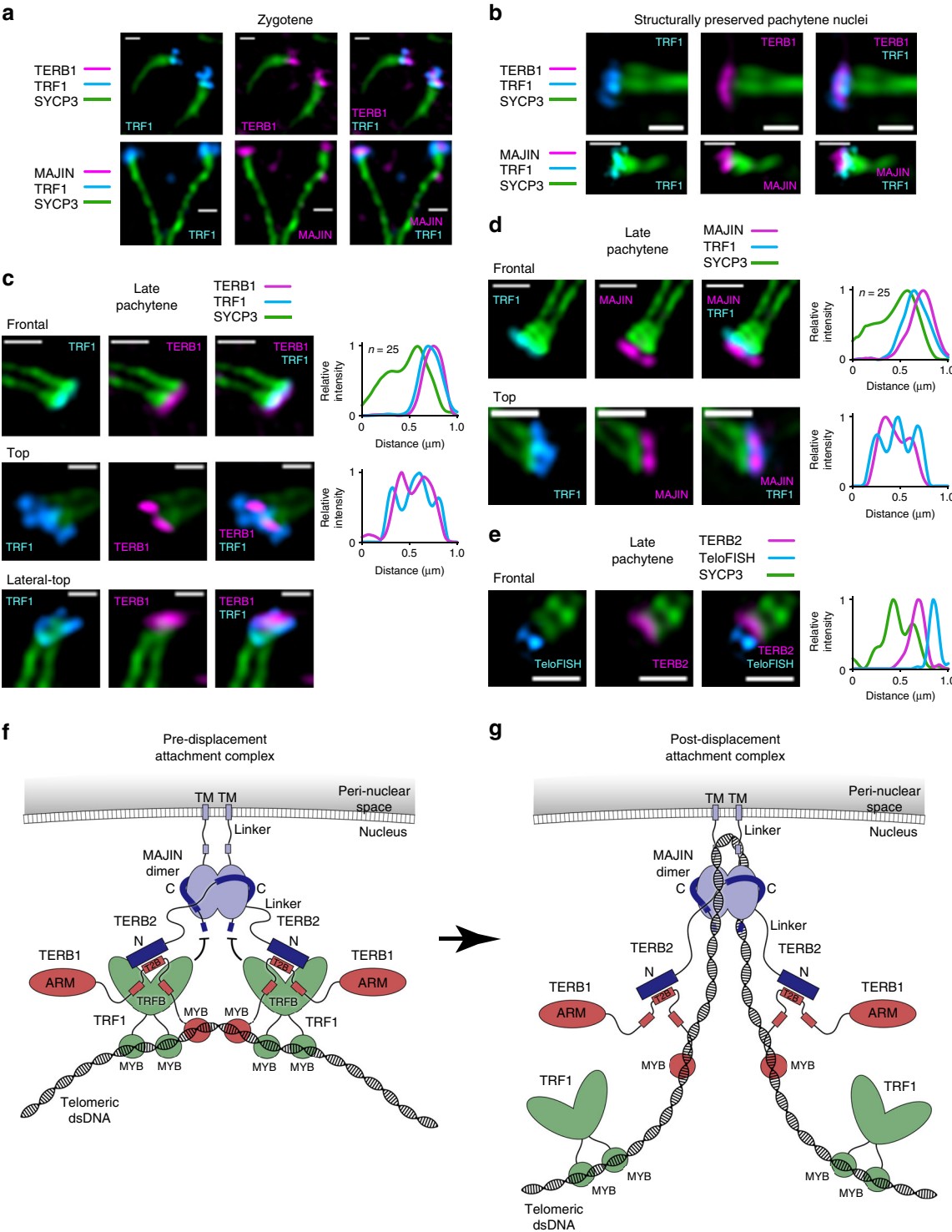

**Fig. 8** TRF1-mediated chromosomal attachment to the meiotic telomere complex. **a–e** Structured illumination microscopy. Wide-field images, plot analyses and additional orientations are shown in Supplementary Figs. 14 and 15. **a** Spread mouse zygotene spermatocyte chromosomes stained with anti-SYCP3 (green), anti-TRF1 (cyan) and anti-TERB1 (top) or anti-MAJIN (bottom) (magenta). Scale bars, 0.3 μm. **b** Structurally preserved mouse spermatocyte pachytene nuclei stained with anti-SYCP3 (green), anti-TRF1 (cyan) and anti-TERB1 (top) or anti-MAJIN (bottom) (magenta). Scale bars, 0.3 μm. **c–e** Spread mouse pachytene spermatocyte chromosomes stained with anti-SYCP3 (green) and (**c**) anti-TERB1 (magenta) and anti-TRF1 (cyan), (**d**) anti-MAJIN (magenta) and anti-TRF1 (cyan), and (**e**) anti-TERB2 (magenta) in combination with telomere fluorescence in situ hybridisation (cyan; TeloFISH). Scale bars, 0.3 μm and 0.5 μm. Normalised intensity-distance plots are shown; frontal plots represent averages of multiple images (n = 25 telomeres), other orientation plots represent individual images. Source data are provided as a Source Data file. **f, g** Model of meiotic telomere attachment to the nuclear envelope. **f** Telomere ends are initially recruited through TRF1, which interacts with the meiotic telomere complex via TERB1, mediating its indirect interaction with telomeres. **g** TRF1 is subsequently displaced, allowing the meiotic telomere complex to bind directly to telomeric DNA, with TRF1 associating with surrounding telomeric DNA. Source data

experimental map was generated by PHENIX Autosol[62] utilising the combined unmerged intensity data. An initial model was built by placing helices and strands by PHENIX Autobuild[62], and was extended and completed through iterative automated and manual re-building in Coot[63]. The structure was refined using PHENIX refine[62], using isotropic atomic displacement parameters with eighteen TLS groups. The structure was refined against 2.90 Å data, to $R$ and $R_{free}$ values of 0.2542 and 0.3039 respectively, with 98.40% of residues within the favoured regions of the Ramachandran plot (0 outliers), clashscore of 0.46 and overall MolProbity score of 0.66.

**Structure solution of MAJIN$_{1-106}$–TERB2$_{168-207}$.** MAJIN$_{1-106}$–TERB2$_{168-207}$ protein crystals were obtained through vapour diffusion in sitting drops, by mixing 100 nl of protein at 30 mg ml$^{-1}$ with 100 nl of crystallisation solution (0.12 M 1,6-hexanediol; 0.12 M 1-butanol; 0.12 M 1,2-propanediol (racemic); 0.12 M 2-propanol; 0.12 M 1,4-butanediol; 0.12 M 1,3-propanediol, 39.1 mM bicine pH 5.03, 60.9 mM Trizma pH 10.83; 12.5% w/v PEG 1000; 12.5% w/v PEG 3350; 12.5% v/v MPD) and equilibrating at 20 °C. Crystals grew overnight, were harvested after 1 week and flash frozen in liquid nitrogen. X-ray diffraction data were collected at 0.9763 Å, 100 K, as 2000 consecutive 0.10° frames of 0.050 s exposure on a Pilatus 6 M detector at beamline I03 of the Diamond Light Source synchrotron facility (Oxfordshire, UK). Data were indexed and integrated in XDS[59], scaled in XSCALE[60] and merged in Aimless[61]. Crystals belong to orthorhombic spacegroup C222$_1$ (cell dimensions a = 59.97 Å, b = 88.39 Å, c = 111.67 Å, α = 90°, β = 90°, γ = 90°), with a 2:2 MAJIN–TERB2 heterotetramer in the asymmetric unit. Structure solution was achieved through molecular replacement using a single MAJIN chain of the selenomethionine derivative SAD structure as a search model, using PHASER[64]. Model building was performed through iterative re-building by PHENIX Autobuild[62] and manual building in Coot[63]. The structure was refined using PHENIX refine[62], using isotropic atomic displacement parameters with three TLS groups per chain. The structure was refined against data to 1.85 Å resolution, to $R$ and $R_{free}$ values of 0.1883 and 0.2072 respectively, with 98.49% of residues within the favoured regions of the Ramachandran plot (0 outliers), clashscore of 0.22 and overall MolProbity score of 0.58.

**Meiotic spreads and nuclei preservation.** Primary antibodies used in this study were as follows: rabbit antibodies against mouse TRF1 (Alpha Diagnostic; TRF12-A, 1:100), TERB2 (1:100);[42] mouse against mouse SYCP3 (Abcam; ab 97672, 1:150); guinea pig against C-terminus rat SYCP2 (Seqlab; Göttingen, Germany; 1:100), against 13-aa (HAGPNVYKFIRYGN) from mouse N-terminus MAJIN (Seqlab; Göttingen, Germany; 1:20), against 15-aa (LDKEKTFDQKDSVSQ) from mouse C-terminus TERB1 (Seqlab; Göttingen, Germany; 1:20) and against 103-aa from mouse C-terminus TERB1 (1:200)[65]. Meiotic cell spreading and structural preservation of meiotic nuclei were performed as previously described[66,67], with minor modifications. Wild-type mice strain C57BL/6 J (30-day-old) were anesthetized and killed with $CO_2$. For chromosome spreading, after decapsulation of mice testes, seminiferous tubules were suspended in hypotonic buffer (30 mM Tris-HCL, 17 mM trisodium citrate, 5 mM EDTA, 50 mM sucrose, 5 mM dithiothreitol, pH 8.2) for 30 min. The resultant swollen tubules were transferred into a drop of 100 mM sucrose on a new slide. The tubules were desegregated with two fine forceps until cells were suitably suspended. Once a cell suspension was achieved, a Superfrost Plus™ slide (ThermoFisher Scientific) was immediately dipped into 1% (w/v) paraformaldehyde (PFA), pH 9.2, 0.15% (v/v) Triton X-100. After allowing excess solution to drip away, the last droplet was kept in one corner of the slide and the cell suspension was placed into this droplet. In the preparation of structurally preserved spermatocyte nuclei, after the decapsulation of mice testes, seminiferous tubules were disrupted with two razor blades until a cell suspension was achieved in cold PBS (140 mM NaCl, 2.6 mM KCl, 6.4 mM Na$_2$HPO$_4$, 1.4 mM KH$_2$PO$_4$, pH 7.4). The suspension was filtered through a nylon filter (mesh size 25–30 μm) and subsequently centrifuged for 10 min at 500 $g$ at 4 °C. The pellet was resuspended and subsequently incubated in eight-well Nunc™ Lab-Tek™ II Chamber Slide™ System (Thermo Fischer Scientific) for 1 h at room temperature. During this time, cells sank to the bottom of the chambers and adhered to the surface due to previous treatment of the chambers well with 0.01% poly-L-lysine in ddH$_2$O (1 h at room temperature). Cells were fixed with 1% paraformaldehyde in ddH$_2$O for 5 min at room temperature. Subsequently, cells were permeabilized with 0.1% Triton-X-100 solution, for 10 min at room temperature. Immediately, cells were blocked with 5% milk powder 5% FCS in PBS, pH 7.4 for > 30 min and then incubated with primary antibodies at room temperature for > 1 h. Post-incubation, samples were washed (3 × 5 min) in PBS (0.13 M NaCl, 0.0027 M KCl, 0.01 Na$_2$HPO$_4$, 0.018 M KH$_2$PO$_4$, pH 7.4). Prior to a 30 min secondary antibody incubation, slides were again blocked for 30 min. During the final 10 min of the secondary antibody incubation, a few drops of Hoechst33258 (1:333 in PBS) were added. Samples were again washed (3 × 5 min), incubated with PBS and stored at 4 °C.

**Structured illumination microscopy and data processing.** Image acquisition was performed as previously described[68]. 2D and 3D SIM images of prophase I stages were scanned using Zeiss Elyra S.1 (Carl Zeiss Microscopy GmbH) equipped with a 63x oil/1.4 oil Plan-Apochromat DICM27 and 63 × /1.2 W KorrM27 objective DICIII. The system also included an X-cite (LED) illumination lamp; 405, 488, 592

and 647 nm lasers with five grid rotation, five5 shifts and PCO Edge 5.5 sCMOS camera. Zeiss immersion oils ranging in refractive index from 1.33 (water) to 1.518 were used, depending on the objective and sample conditions analysed. Colour channels were carefully aligned using 200 nm TetraSpeck™ microspheres (Thermo Fischer Scientific, catalogue number: T-7280). Each channel alignment was processed with the available SIM algorithms in the custom software. Image reconstruction was performed using the commercial software ZEN 2012 package (Carl Zeiss Microscopy GmbH) based on previously developed structured illumination algorithms[69]. Images were acquired in the form of Z-stacks. Sections were acquired at 0.1 μm. The range used for the structurally preserved nuclei was 3 μm and 0.80 μm for chromosomes spreading. Negative values in the reconstruction process were not discarded comprising the full dynamic range and 16-bit colour depth. To analyse signal distribution of MTC proteins, late pachytene-spermatocytes were selected by the marker pair XY chromosomes[70]. After maximum intensity projection for each image were processed, background was subtracted by measuring the average intensity in an ROI position outside the sample area and subtracting this fixed intensity value from each in the image equally. Quantification of signal distribution was measured in a constant region with fix length (1 μm) lines of interested regions by recording the intensity profile for each channel in Fiji[71]. Intensity profiles were normalized to their respective maximum intensity.

**Telomere fluorescence in situ hybridization.** Telomere fluorescence in situ hybridization (TeloFISH) was combined with immunofluorescence as described previously[67], with minor modifications. Testis cell spreads were dehydrated in an alcohol series (70, 80 and 100%) for 5 min. The hybridization solution and both labelling reactions were pre-heated to 95 °C for ~15 min. Immediately before hybridization, 6 μl of each of the two labelling reactions was added to 80–100 μL of the pre-heated hybridization solution mix, and pipetted onto the slides in a moisture chamber. Note, we found it beneficial to mark a central region on the slide using a grease pencil to ensure the same region is subjected to the following immunofluorescence procedure. The chamber was tightly sealed with parafilm to avoid evaporation. The chamber was immediately placed into an oven pre-heated to 95 °C and denatured for 20 min. We further note that working quickly and smoothly during this procedure greatly increases the success of the protocol. After rinsing for 10 min, mice testis cells were spread in 2x SCC (SSC 0.3 M NaCl, 0.03 M Na-citrate; pH 7.4) and denatured at 95 °C for 20 min, and placed in 40 μl of hybridization solution (30% formamide, 10% dextran sulphate, 250 μg ml$^{-1}$ E. coli DNA in 2 × SSC) supplemented with 10 pmol digoxigenin-labelled (TTAGGG)$_7$/ (CCCTAA)$_7$ oligomers. Hybridization was performed at 37 °C overnight in a humid chamber. Slides were washed two times in 2 × SSC at 37 °C for 10 min each and blocked with 0.5% blocking-reagent (Roche, Mannheim, Germany) in TBS (150 mM NaCl, 10 mM Tris/HCl; pH 7.4). Samples were incubated with mouse anti-digoxigenin antibodies (Roche, Mannheim, Germany; 1:50) according to the manufacturer's protocol and bound antibodies detected with Alexa488 (Thermo scientific; 1:200) anti-mouse secondary antibodies. Following the TeloFISH procedure, samples were prepared for immunofluorescence by blocking with 5% (w/v) milk, 5% (v/v) FCS in PBS, pH 7.4. Slides were incubated with the first primary antibody overnight, washed two times in PBS for 10 min each and incubated with the corresponding secondary antibody as described above. Finally, slides were again washed in PBS before incubating with the second primary antibody. After further washing in PBS, samples were exposed to the corresponding secondary antibodies.

**Protein sequence and structure analysis.** The MAJIN sequence and boundaries utilised in this study relate to the human MAJIN isoform X1 (254 amino acids; accession number XP_024304215.1), which was chosen as it contains all necessary domains and most closely matches the canonical isoforms conserved across mammals. TERB1, TERB2 and TRF1 sequences relate to their canonical human isoforms (accession numbers Q8NA31, Q8NHR7 and P54274, respectively). Amino acid conservations were calculated for full length proteins and the MAJIN$_{Core}$–TERB2$_C$ crystal structure by ConSurf (http://consurf.tau.ac.il/). Structural assemblies were analysed by PISA. Molecular structures images were generated using the PyMOL Molecular Graphics System, Version 2.0 Schrödinger, LLC.

**Ethics statement.** Animal care and experiments were conducted in accordance with the guidelines provided by the German Animal Welfare Act (German Ministry of Agriculture, Health and Economic Cooperation). Animal housing and breeding was approved by the regulatory agency of the city of Würzburg (Reference ABD/OA/Tr; according to §11/1 No. 1 of the German Animal Welfare Act).

## Data availability

Crystallographic structure factors and atomic co-ordinates have been deposited in the Protein Data Bank (PDB) under accession numbers 6GNX and 6GNY. A Reporting Summary for this Article is available as a Supplementary Information file. The Source Data underlying Figs. 4c,d, 5a,f, 6a, 7b–d, f–k and 8c–e and Supplementary Figs. 1e–g, 4f, g, 5a, c, d, 6a, b, 10a–c, 11a, b, 12a, b, d–f and 13g, h are provided as a Source Data file. All other data are available from the corresponding authors upon reasonable request.

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

## Acknowledgements

We thank Diamond Light Source and the staff of beamlines I03, I04, I04-1 and B21 (proposals mx13587, mx18598 and sm15836). We thank S. Korolchuk and J.A. Endicott for kindly providing purified phosphorylated CDK1-CyclinB, A. Basle and H. Waller for assistance with X-ray crystallographic and CD data collection, H. Shibuya and A. Tóth for kindly providing TERB2 and TERB1 antibodies, and D. Jain for advice on the paper. I.d.C. is supported by a DAAD fellowship (Germany). R.B. is supported by a German Science Foundation grant (grant number Be 1168/8-1). O.R.D. is a Sir Henry Dale Fellow jointly funded by the Wellcome Trust and Royal Society (grant number 104158/Z/14/Z) and is also supported by a Royal Society Research Grant (Grant Number RG170118).

## Author contributions

M.G. crystallised the MAJIN$_{Core}$–TERB2$_C$ selenomethionine derivative. J.M.D. crystallised truncated MAJIN$_{Core}$–TERB2$_C$. A.E.M. and J.M.D. performed biochemical and biophysical studies on MAJIN–TERB2–TERB1 and TRF1–TERB1–TERB2 complexes. J. M.D. and M.G. performed biophysical experiments on MAJIN–TERB2 and all DNA-binding experiments. J.M.D. analysed DNA-binding data and performed phosphorylation experiments. O.R.D. analysed SAXS data and solved the MAJIN–TERB2 crystal structures. I.d.C. performed structured illumination microscopy experiments. L.T.S performed initial experiments. J.M.D., A.E.M, R.B. and O.R.D. designed experiments. O. R.D. wrote the paper.

## Additional information

**Competing interests:** The authors declare no competing interests.

