## [Peer Review File · Nature Communications]

Reviewers' comments:

Reviewer #1 (Remarks to the Author):

Meiosis presents chromosome topological challenges at the time in which homologous chromosomes pair, and recombine. Failure in these fundamental processes results in phenotypes ranging from sterility to aneuploid offspring with developmental abnormalities. Thus, work like this focusing in understanding basic mechanisms promoting proper chromosome associations, are very important for the advancement of the field.

Chromosome interactions and meiotic recombination in prophase are invariably accompanied by prominent telomere-led chromosome movements or rapid prophase movements (RPMs) that are widely conserved in organisms from yeast to mammals. In all organisms studied so far meiotic prophase chromosome movements are driven by the connection of telomeres to the cytoskeleton via protein bridges through the intact nuclear envelope termed the LINC complex. The LINC consists of inner nuclear envelope SUN domain proteins and outer nuclear membrane KASH domain proteins. Telomeres associate with the intranuclear domain of the SUN proteins, and the extranuclear portion of the KASH protein connects to the cytoskeleton. Using cellular and genetic approaches previous works have shown that interaction of the meiotic telomeres with the nuclear envelope drives RPMs. Importantly, they identified the proteins (TRF1, TERB1, TERB2, and MAJIN) and proposed a mechanism explaining meiotic telomere-nuclear membrane interaction. However, the architecture of this protein complex is not well understood, and advances in the molecular structure of this complex are necessary to explain functional and regulatory mechanisms for telomere engagement with other intermembrane and cytoskeleton proteins and mechanisms.

In this work Duncie et. al., present: (1) the crystal structure of MAJIN-TERB2, which provides maximum resolution structure for a fundamental part of the structure linking telomere and nuclear membrane; (2) light and X-ray scattering studies of wider complexes, which together with (3) biochemical evidence for a MAJIN-TERB2 hetero-tetramer binding to DNA, and (4) that MAJIN-TERB2 is tethered through long flexible linkers to the inner nuclear membrane and two TRF1-binding 1:1 TERB2-TERB1 complexes, present strong evidence for a dynamic functional model for telomere attachment to the nuclear membrane. Finally, results obtained with imaging on spermatocytes and biochemical approaches are used to propose a model for telomere attachment in which MAJIN-TERB2-TERB1 recruits telomere-bound TRF1, which is later displaced, to allow MAJIN-TERB2-TERB1 complex binding to telomeric DNA and form a mature attachment plate.

My overall appreciation is that this work is outstanding. The quality of results is excellent, and very well designed experiments clearly support this highly interesting work. The results obtained will be of interest for a wide range of researches in a number of fields such as meiosis, nuclear biology, cellular and membrane biology, and nuclear structure. Although the technical nature of a good fraction of the work, the manuscript is clearly written and discussed.

I think this work is well suited for a high impact journal such as Nature Communications. I have only minor comments and advice for improvement in the written and experimental sections:

1- In the Abstract and in the Introduction it is stated that rapid prophase movements facilitate homologous interactions. For example in the abstract: "Meiotic chromosomes undergo rapid prophase movements to facilitate the formation of inter- homologue recombination intermediates that underlie synapsis, crossing over and segregation." I think the authors should be carefully with this concept. Although this is general accepted, specific work demonstrating this fact, regarding the importance of rapid prophase movements in promoting homologous pairing, is rather scarce (actually only suggestive in mammals). So, the authors should caveat more.

In line 59 this concept has been more carefully stated: "are thought to facilitate the homology search by driving interactions between chromosomes and disrupting non-homologous contacts." Also, they should clearly state the few reports supporting this view. For example: in yeast, Lee CY,

Conrad MN, Dresser ME. PLoS Genet. 2012;8(5):e1002730. In *C. elegans*: Wynne DJ, Rog O, Carlton PM, Dernburg AF. J Cell Biol. 2012.

2- In the same caliber: "extensive recombination-mediated homology searches throughout the genome". While this is clearly a plausible possibility, in an alternative view, it has been shown that chromosomes with substantial homology are in near proximity in pre-meiotic stages, which may already reduce the area of homology search.

I note that the concerns I raise in points 1 and 2 are minor. The authors use these concepts only to introduce the work. Still, I think clarification may be required in view of readers that are not aficionados.

3- Please clarify "an array of recombination intermediates".

4- Results: is it not clear to me why an extensive basic patch on the surface of each protomer, formed of highly conserved amino acids in the grasping β -sheet of MAJIN, suggest a direct interaction with DNA. Has this been shown for other proteins before?

5- Figures in which DNA binding is performed: How many times the DNA binding assay was repeated? Please show quantification and introduce error bars in the corresponding graph. This may be especially important to compare different conditions.

6- Regarding spermatocyte imaging in Fig. 5 and Fig. 8. Likely, native structures are lost in the preparation of spermatocyte spreads. It is recommended the use of preparations that likely conserve nuclear and cytoplasmic 3D structure, such as seminiferous tubule squashes. I recommend confirming key findings in Fig. 5 and 8 by repeating immunostaining on spermatocyte preparations that keep nuclear 3D structure.

7- Are there differences in the architecture of the complexes at telomere ends between imaged pachytene and zygotene chromosomes? I think this should be noticed.

Reviewer #2 (Remarks to the Author):

In this work, Duncanson, Milburn, Gurusaran et al use a multitude of experimental methods including X-ray crystallography, SEC-MALS, EMSAs, and structured illumination microscopy to provide insights into the structure, stoichiometry, and DNA-binding properties of the MAJIN-TERB2-TERB1-TRF1 complex that attaches telomeres to the INM during meiosis. They solve the crystal structure of a complex containing the interacting regions of TERB2 and MAJIN to reveal a 2:2 composition containing 2 lobes of MAJIN wrapped around by two TERB2 fragments that adopt greatly extended conformations. The structure reveals how TERB2 participates in MAJIN homodimerization and stabilization; and suggests a basic patch on each MAJIN monomer on opposite faces of the homodimer that may be important for DNA binding and looping. The 2:2 stoichiometry is further verified using SEC-MALS and can be compromised by mutations in MAJIN (F73E-Y75E) that lie on the MAJIN dimerization interface. The authors demonstrate DNA binding of TERB2-MAJIN complexes that do not discriminate between double-stranded/single-stranded DNA or telomeric/non-telomeric DNA substrates. The DNA binding was shown to rely on basic patches in linker/unstructured regions in MAJIN and TERB2 as removal of these regions were associated with weaker binding. Dimerization of MAJIN also had a negative influence on DNA binding. DNA binding was further demonstrated with EM experiments that showed circular TERB2-MAJIN structures colocalize on DNA "strings". The authors then build on the MAJIN-TERB2 core to characterize the MAJIN-TERB2-TERB1 complex using SEC-MALS and SAXS and ab initio modeling (in addition to their crystal structure) and find that MAJIN-TERB2-TERB1 exists 2:2:2, while TERB2-TERB1 exists 1:1. Using structured illumination microscopy they show that TERB2 is proximal to the chromosomal axis compared to MAJIN, which seems more distal from the chromosome axis. The authors then include TRF1 (the TRFH domain) into the complex to show that the TRF1-TERB1-TERB2 stoichiometry is 2:1:1, while TRF1:TERB1 stoichiometry is 2:1. They also build a model for this complex using SAXS ab initio methods. The authors then return to EMSAs to show that while TRF1 (TRFH) displaces MAJIN-TERB2 off DNA, the reverse is not true – full length TRF1 bound to DNA can recruit MAJIN-TERB2-TERB1. Finally TRF1 was shown to be more proximal to the

chromosome axis than TERB1 or MAJIN while telomeric DNA (by FISH) was most distal. The greatest strengths of this work include the novel structure of the MAJIN-TERB2 interface, and the very impressive reconstitution and determination of stoichiometry of the various meiotic telomere complexes. However there are multiple disagreements between the current and published work that are pertinent to some of the major conclusions of the study. Furthermore the DNA binding studies need to be further strengthened to bolster some of the conclusions.

1. To make a strong argument about there being no discrimination for double vs. single stranded or telomeric vs. telomeric DNA, EMSA should be done under conditions where the $[DNA] \ll K_d$ of the interaction. Also, as the authors are reporting a new protein-DNA interaction, it seems important to determine the absolute strength (K_d) of the interaction. For example, telomeric proteins like TRF1 and TRF2 are known to bind telomeric DNA with high affinity (nanomolar regime). Is the MAJIN-TERB2-DNA interaction weaker, and if so, how weak? If it indeed is in the micromolar regime – as suggested by crude inspection of the raw EMSA data, do the authors believe it is physiologically relevant?

2. The authors show that unstructured/extended basic regions of both MAJIN and TERB2 are important for DNA binding. However neither of these regions are resolved in the crystal structure. As they propose a model where the DNA loops around MAJIN dimers along the two basic surfaces on the MAJIN lobes, the authors should validate this using mutations of basic residues in the MAJIN lobes. If the lobe mutants don't affect DNA binding, it would suggest that a) the DNA looping model needs to be revised, and b) all the DNA binding of MAJIN-TERB2 comes from basic linker/extended regions.

3. There is one major concern in the SEC-MALS experiment, which is in regard to the oligomeric status of TERB1 and TERB1-TERB2. The authors suggest that TERB1-TERB2 is a 1:1 complex. This goes against biochemical studies by Shibuya et al (Cell), which show that TERB1 in TERB1-C (which is TRFB+myb) as well as TERB1-C-TERB2 is homodimeric by gel filtration analysis, and that MBP-TERB1(TRFB) pulls down His-TERB1(TRFB) (see Supp fig 5 of that paper). There are at least two explanations for this discrepancy. Firstly, it is possible that the differences in techniques (SEC-MALS vs gel filtration alone or MBP pull down) are responsible for the incongruent observations. Secondly, the construct for TRFB used in the current study by Duce et al is much shorter than the construct from Shibuya et al (starting at aa 561 as opposed to starting at aa 523). As stoichiometry determination is a major strength of the current study, it is important for the authors to resolve this difference. One direct route to resolve the disparate observations would be to perform SEC-MALS with TERB1(TRFB) as well as TERB1(TRFB)-TERB2 starting at TERB1 aa 523 rather than 561.

4. The authors claim that TRF1:TERB1 is 2:1 implying that only one site of a TRF1 dimer is bound by TERB1. This has been shown previously by Pendlebury et al (NSMB; figure 4) and is therefore not a new observation.

5. In light of the model for the full DNA-TRF1---MAJIN-INM complex proposed here, it is surprising how the binding of TRFH to TERB1 inhibits binding of DNA to TERB2-MAJIN. As long linkers are separating the TRF1-TERB1 interface from the TERB2-MAJIN-DNA interface how do the authors invoke this inhibition is taking place? The binding data shown for TERB2-TRF1 direct binding is very weak (Sup Figure 9E) and seems insufficient to explain the EMSA results. It is known that TRF1 has a very acidic N-terminus and so it is not surprising that it can associate simply by electrostatics to a positively charged surface such as the basic C-terminus of TERB2 (as shown in the supplement). However, it is difficult to argue based on this that the TERB2-TRF1 interaction is either specific or has implications for DNA-binding. Finally, if the authors propose that TRF1 pushes TERB2-MAJIN off DNA, that would suggest a very weak interaction of TERB2-MAJIN with DNA as it can be disrupted by addition of a protein (K_d determination with DNA as suggested above should be able to address this).

6. Three different groups have shown that the phosphomimetic mutation of TERB1 T648 breaks the TERB1-TRF1 interaction (Shibuya et al; NCB, Pendlebury et al; NSMB; Long et al NSMB). The TERB1 constructs in these studies span the TBM and even full length TERB1, and employ different experimental methods (co-IP, fluorescence flow cytometry, yeast two hybrid, immunofluorescence in human cells, immunofluorescence in mouse spermatocytes). And in all cases the T648E/D mutation greatly reduces binding to TRF1. However, in the current study, this mutation doesn't affect TRF1 binding. One reason for this discrepancy could be the concentration at which the SEC studies were performed. Protein concentrations of 10 mg/ml were used (at least in SEC-MALS), which amounts to ~mM concentrations of TERB1 (TRFB). As these concentrations are much higher than the reported Kd for the TERB1-TRF1 interaction (from the two NSMB studies), it seems highly likely that this is the reason why the authors do not see an effect of the T-to-E mutation. Direct Kd measurements need to be performed to substantiate the authors claim that there is no effect of the phosphomimetic mutation on the interaction. The TERB1 and TRF1 concentrations in vivo are probably much lower than mM, and hence the physiological TRF1-TERB1 interaction would be expected to still be sensitive to this mutation as has been already demonstrated by all three studies referenced above.

7. In all of the SIM experiments done in mouse spermatocytes, the sub-stage of pachytene is not indicated. The colocalization of TERB1-TERB2-MAJIN with shelterin (including TRF1) and telomeric DNA depends on the stage in pachytene (Shibuya et al; Cell). In early pachytene all these proteins colocalize, while later in pachytene the signals of these complexes separate. In the SIM results shown in the current study the signals of TRF1 seem partially separated (and chromosome axis proximal) from that of TERB1-TERB2-MAJIN. Does this mean that all of the images captured represent late pachytene? If so, a marker for late pachytene should be included. If the images have not been sorted for a particular stage in pachytene, it is surprising that TRF1 has "detached" from the telomeric signal so early; this would not follow the imaging studies from the Watanabe lab on cap exchange. On a related note, some of the profile changes in the SIM experiments seem subtle and therefore inclusion of statistical criteria (standard deviations, p-value, etc) seems important.

8. The location of the MAJIN basic patch linkers towards the membrane suggests a role for them associating with the inner nuclear membrane containing phospholipids. The DNA binding activity could be an artifact of the absence of any membrane component in the experiment. Although it is difficult to adapt the EMSA to address this, the authors could perform INM-tethering experiments as performed previously by the Lei and Watanabe labs with BP1/BP2 mutation constructs to ask if these disrupt membrane binding.

Minor points:

1. The method of expression and purification, and the quality of the purified full length TRF1 protein should be included as very few biochemical studies have been performed with purified full-length TRF1 presumably because it is a difficult protein to study biochemically.
2. It is not clear how the SAXS data are used in the CORAL or ab initio modeled structures. There doesn't seem to be any SAXS envelope suggestive of these linkers.
3. The number of replicates "n" is not indicated in the quantitations in figure 8 b, c, d. If these are for representative telomeres, then replicates should be quantified so that at least tens of foci are quantified.
4. A membrane stain (if this is feasible) would be very useful in the SIM experiments so that one can calibrate the distance from both the chromosomal and membrane directions in each panel.
5. TIN1 should be changed to TIN2 in line 88.

Reviewer #3 (Remarks to the Author):

The authors present an impressively extensive study on the MAJIN-TERB2-TERB1 complex. Utilizing X-ray crystallography combined with solution scattering and biochemical techniques, they seek to solve the structure of this complex in solution and give insights into the telomere attachment mechanism and subsequent manipulation.

This reviewer has experience primarily in the small-angle x-ray scattering (SAXS) aspect of the study and therefore will primarily comment on this aspect of the study. In short, while it is possible that the SAXS data are sufficient, the data are not presented with enough clarity and controls to ensure that the reported conclusions are accurate and represented by the data. In this reviewer's opinion, the paper could be published only if these concerns are addressed through additional analysis being presented.

Overarching comments:

1. There should be a concentration series to control for common recurring issues in SAXS.

a. The first is to control for aggregation. Aggregation effects have been attempted to be controlled through SEC-SAXS methods, but these are insufficient. They do not rule out effects due to time (the molecules spend a significant amount of time on a molecular time scale outside of the separation column) or radiation-induced aggregation effects. The currently reported flow rate (0.5ml/min or ~8ul/s) means that 8ul of sample are being exposed to 1 second of up to 10^{12} photons per second (as the authors do not report a beam strength, the reviewer assumes the full beam available at B21 of the Diamond Light Source was used). This flux is more than sufficient to cause radiation damage in certain proteins.

b. The second possible issue is interparticle interference. Due to running at high concentrations the experiments in this study are especially likely to encounter these effects. Both interparticle interference and aggregation can have large effects at the lowest q values, which affect the measurement of the largest dimensions of the sample.

2. Concentration series are harder on SEC-SAXS systems. Barring the possibility to do these, extensive Guinier analyses should be completed. However, at no point in the analysis do the authors present their Guinier regimes or their R_g values. These should both be presented at least in the supplementary information. Additionally, for all $P(r)$ determinations, the authors should ensure that the R_g calculations roughly match the Guinier analysis.

3. A constant issue with SEC-SAXS is buffer subtraction, due to the often long times between sample and background buffer measurements. The authors should explain how careful background subtraction was achieved. Additionally, as mentioned above, a transparent Guinier analysis would allow the reader to evaluate the quality of the buffer subtraction.

To see the possible pitfalls of SEC-SAXS buffer subtraction and proposed fixes, please see the work of Malaby et al.[1]

4. D_{max} is an important and difficult parameter to determine when calculating $P(r)$. There are numerous instances in the manuscript where the utilized D_{max} values are suspect. $P(r)$ profiles should normally approach zero naturally at D_{max} in a concave manner.[2] There are frequent times in the manuscript where the authors have likely chosen an artificially small D_{max} value.

While this seems like a small thing, the D_{max} value has a strong effect on the $P(r)$ profile and the consequent ab initio reconstructions.

5. For all SAXS profiles, the authors should include y-axis units. If the units are “arbitrary” they should be indicated as such.

6. There are two excellent articles the author suggests for handling of SAXS data and Dmax determination by Skou et al.[3] and the aforementioned Jaques et al.[2]

Specific comments:

1. Figure 3b: As said in the general comments, one would need to see a Guinier analysis to determine whether or not the experimental data is adequate. In particular, the reviewer is worried about the very low q data for the top SAXS profile.

2. Figure 3c: The dotted P(r) profile (MAJINCore-TERB2C) likely has an artificially low Dmax which is what leads to the “shoulder” around 100 angstroms. The solid, black line (MAJINCoreTr-TERB2C-Tr) also may have a slightly artificially truncated Dmax.

3. Figure 4d: The reviewer was confused by the bottom-right figure. Should the reviewer be seeing anything in this picture? If not, it would be helpful to include a statement explaining what it seeks to display.

4. Figure 5d: The top SAXS profile (TERB2-TERB1) does not go to the same low q value as all of the other data. Is there a reason for this? The reviewer is concerned that there were problems with this low-q data and this is why it was not included.

5. Figure 5d: All other Crysol fits in the manuscript go to zero. The “CORAL model” theory curve (gray curve) does not go to q=0. It is also truncated at high-q (see, for comparison, see the profile directly above it in red.) These points of the fit should be included so that the reader can accurately evaluate the model.

6. SupplementalFigure5e: Again, these Dmax values seem problematic. Especially worrisome is the TERB2N-TERB1T2B profile that is subsequently used to create the ab initio model in 5f.

7. Figure 6c: There seem to be subtraction issues for the TERB2-TERB1 and TRF1-TERB1-TERB2 scattering profiles. In the former, the lowest q data seems to increase; in the latter, the lowest q data rapidly decreases. Again, a Guinier analysis would make these issues clearer and possibly allow the authors to obtain a better analysis.

8. Figure6d: The authors should show at least one rotation of the final structure to evaluate whether the crystal structure fits into the ab initio model in the third (into/out of page) dimension.

9. Supplemental Figure 8e: There seems to be an issue with two of the Dmax values in the P(r) profile. The TRF1-TERB1-TERB2 profile has a well-determined Dmax value as indicated by the slope of the profile whilst approaching Dmax. The other two have more abrupt terminations. The reviewer also wonders why the TERB2N-TERB1T2B P(r) profile matching the scattering profile in Figure 6c was not included?

References:

1. Malaby, A. W. et al. Methods for analysis of size-exclusion chromatography–small-angle X-ray scattering and reconstruction of protein scattering. *J. Appl. Crystallogr.* 48, 1102–1113 (2015).

2. Jacques, D. A. & Trewella, J. Small-angle scattering for structural biology—Expanding the frontier while avoiding the pitfalls. *Protein Sci. Publ. Protein Soc.* 19, 642–657 (2010).

3. Skou, S., Gillilan, R. E. & Ando, N. Synchrotron-based small-angle X-ray scattering (SAXS) of proteins in solution. *Nat. Protoc.* 9, 1727–1739 (2014).

Response to reviewers' comments
'Structural basis of meiotic telomere attachment to the nuclear envelope by
MAJIN-TERB2-TERB1'

We are pleased that the reviewers recognise the interest and importance of this work. Here, we provide responses to the questions and comments raised by the reviewers, and outline how, in accordance with their requests, we have revised the manuscript and added additional experimental data.

Reviewer #1 (Remarks to the Author):

Meiosis presents chromosome topological challenges at the time in which homologous chromosomes pair, and recombine. Failure in these fundamental processes results in phenotypes ranging from sterility to aneuploid offspring with developmental abnormalities. Thus, work like this focusing in understanding basic mechanisms promoting proper chromosome associations, are very important for the advancement of the field.

Chromosome interactions and meiotic recombination in prophase are invariably accompanied by prominent telomere-led chromosome movements or rapid prophase movements (RPMs) that are widely conserved in organisms from yeast to mammals. In all organisms studied so far meiotic prophase chromosome movements are driven by the connection of telomeres to the cytoskeleton via protein bridges through the intact nuclear envelope termed the LINC complex. The LINC consists of inner nuclear envelope SUN domain proteins and outer nuclear membrane KASH domain proteins. Telomeres associate with the intranuclear domain of the SUN proteins, and the extranuclear portion of the KASH protein connects to the cytoskeleton. Using cellular and genetic approaches previous works have shown that interaction of the meiotic telomeres with the nuclear envelope drives RPMs. Importantly, they identified the proteins (TRF1, TERB1, TERB2, and MAJIN) and proposed a mechanism explaining meiotic telomere-nuclear membrane interaction. However, the architecture of this protein complex is not well understood, and advances in the molecular structure of this complex are necessary to explain functional and regulatory mechanisms for telomere engagement with other intermembrane and cytoskeleton proteins and mechanisms.

In this work Duncce et. al., present: (1) the crystal structure of MAJIN-TERB2, which provides maximum resolution structure for a fundamental part of the structure linking telomere and nuclear membrane; (2) light and X-ray scattering studies of wider complexes, which together with (3) biochemical evidence for a MAJIN-TERB2 hetero-tetramer binding to DNA, and (4) that MAJIN-TERB2 is tethered through long flexible linkers to the inner nuclear membrane and two TRF1-binding 1:1 TERB2-TERB1 complexes, present strong evidence for a dynamic functional model for telomere attachment to the nuclear membrane. Finally, results obtained with imaging on spermatocytes and biochemical approaches are used to propose a model for telomere attachment in which MAJIN-TERB2-TERB1 recruits telomere-bound TRF1, which is later displaced, to allow MAJIN-TERB2-TERB1 complex binding to telomeric DNA and form a mature attachment plate.

My overall appreciation is that this work is outstanding. The quality of results is excellent, and very well designed experiments clearly support this highly interesting work. The results obtained will be of interest for a wide range of researches in a number of fields such as meiosis, nuclear biology, cellular and membrane biology, and nuclear structure. Although the technical nature of a good fraction of the work, the manuscript is clearly written and discussed.

I think this work is well suited for a high impact journal such as Nature Communications. I have only minor comments and advice for improvement in the written and experimental sections:

1- In the Abstract and in the Introduction it is stated that rapid prophase movements facilitate homologous interactions. For example in the abstract: “Meiotic chromosomes undergo rapid prophase movements to facilitate the formation of inter-^[L]_[SEP]homologue recombination intermediates that underlie synapsis, crossing over and segregation.” I think the authors should be carefully with this concept. Although this is general accepted, specific work demonstrating this fact, regarding the importance of rapid prophase movements in promoting homologous pairing, is rather scarce (actually only suggestive in mammals). So, the authors should caveat more.

In line 59 this concept has been more carefully stated: “are thought to facilitate the homology search by driving interactions between chromosomes and disrupting non-homologous contacts.”

Also, they should clearly state the few reports supporting this view. For example: in yeast, Lee CY, Conrad MN, Dresser ME. PLoS Genet. 2012;8(5):e1002730. In C. elegans: Wynne DJ, Rog O, Carlton PM, Dernburg AF. J Cell Biol. 2012.

1. We agree with the reviewer that this issue has not been resolved and so requires appropriate caveats. We have edited the statement in the abstract and figure legend 1a to indicate that RPMs are thought to facilitate these processes (line numbers 25 and 937). We have further added a statement to the introduction citing the yeast and C. elegans studies upon which this is based (line number 59).

2- In the same caliber: “extensive recombination-mediated homology searches throughout the genome”. While this is clearly a plausible possibility, in an alternative view, it has been show that chromosomes with substantial homology are in near proximity in pre-meiotic stages, which may already reduce the area of homology search.

I note that the concerns I rise in points 1 and 2 are minor. They authors use these concepts only to introduce the work. Still, I think clarification may be required in view of readers that are not aficionados.

2. We agree with the reviewer that chromosome topology within the cell likely enhances the chances of homology searches between homologues, thereby reducing the scale of

the search. We have therefore edited the statement to remove the implication that homology searches necessarily extend throughout the genome (line number 53).

3- Please clarify “an array^[SEP] of recombination intermediates”.

3. We are referring to the recombination intermediates that are formed between chromosomes and act as physical tethers. We recognise that the term array may have incorrectly suggested a wider specific structure, and so have changed the text to state that it generates a series of recombination intermediates (line number 53).

4- Results: is it not clear to me why an extensive basic patch on the surface of each protomer, formed of highly conserved amino acids in the grasping β -sheet of MAJIN, suggest a direct interaction with DNA. Has been this show for other proteins before?

4. Clusters of highly conserved basic residues on a protein surface are frequently found on the surface of DNA-binding proteins as they enable electrostatic interactions with the negatively charged phosphodiester backbone. This is particularly the case for proteins that bind without sequence-specificity and has been demonstrated for a large number of protein-DNA interactions *in vitro* and *in vivo* (e.g. histones). We have modified the statement to clarify our rationale (line numbers 176-177). We have not been able to find an example of another β -grasp fold that binds DNA, so this may be a novel function acquired by MAJIN, in accordance with the highly diverse nature of β -grasp domains.

5- Figures in which DNA binding is performed: How many times the DNA binding assay was repeated? Please show quantification and introduce error bars in the corresponding graph. This may be especially important to compare different conditions.

5. The DNA-binding assays are representative of at least three replicates; we have added this statement to the relevant figure legends (line numbers 1013-1014, 1086-1087, 1098, 1189 and 1204). These assays were performed using sufficient DNA to provide clear visual indication of the strength and nature of DNA-binding. In accordance with the requests of reviewer 2 (described in response number 8), we have repeated these analyses at much lower DNA concentrations to ensure they are substantially below the K_D and thereby enable the reliable determination of apparent K_D values. These assays were performed in triplicate, quantified by densitometry and fitted to the Hill equation to determine the apparent K_D . We have added graphs of the data, including error bars and their fits alongside the existing DNA-binding gels (Figures 4d, 7d and 7h; Supplementary Figures 4f and 5d).

6- Regarding spermatocyte imaging in Fig. 5 and Fig. 8. Likely, native structures are lost in the preparation of spermatocyte spreads. It is recommended the use preparations that likely conserve nuclear and cytoplasmic 3D structure, such as seminiferous tubule squashes. I recommend confirming key findings in Fig. 5 and 8 by repeating immunostaining on spermatocyte preparations that keep nuclear 3D structure.

6. In accordance with the reviewer's request, we have performed additional experiments on intact nuclei structures (Figures 5g and 8b; Supplementary Figures 8f and 14c-d). We used fixed whole cells attached to a coverslip instead of squash preparations in order to avoid possible mechanical stress. We used Nunc™ Lab-Tek™ II Chamber Slide™ System (Thermo Fischer Scientific) suitable for SIM as they have a high precision coverslip at the bottom. Also, the sample was adapted to water immersion objective 63x/1.2 W KorrM27 objective DICIII with a higher numerical aperture because the cells are incubated in PBS. These additional analyses confirm our findings in chromosome spreads regarding TERB1-TERB2, TERB1-TRF1 and MAJIN-TRF1 localisation.

7- Are there differences in the architecture of the complexes at telomere ends between imaged pachytene and zygotene chromosomes? I think this should be noticed.

7. We have included additional analyses of zygotene spermatocyte chromosomes (Figures. 5e and 8a; Supplementary Figures 8a and 14a-b). These reveal a similar flanking staining of TERB2 relative to MAJIN, albeit surrounding a single MAJIN focus at a telomere end rather than enclosing the two coalesced foci of paired telomere ends in pachytene (Fig. 5e). We further find partially overlapping distributions of TERB1 and TRF1, and separation of MAJIN and TRF1 along the chromosome axis in zygotene (Fig. 8a), which support our model of MAJIN-TERB2-TERB1 organisation and TRF1 displacement in pachytene. We have added descriptions of these findings to the manuscript (line numbers 238-240 330-335 and 338-340).

Reviewer #2 (Remarks to the Author):

In this work, Dunce, Milburn, Gurusaran et al use a multitude of experimental methods including X-ray crystallography, SEC-MALS, EMSAs, and structured illumination microscopy to provide insights into the structure, stoichiometry, and DNA-binding properties of the MAJIN-TERB2-TERB1-TRF1 complex that attaches telomeres to the INM during meiosis. They solve the crystal structure of a complex containing the interacting regions of TERB2 and MAJIN to reveal a 2:2 composition containing 2 lobes of MAJIN wrapped around by two TERB2 fragments that adopt greatly extended conformations. The structure reveals how TERB2 participates in MAJIN homodimerization and stabilization; and suggests a basic patch on each MAJIN monomer on opposite faces of the homodimer that may be important for DNA binding and looping. The 2:2 stoichiometry is further verified using SEC-MALS and can be compromised by mutations in MAJIN (F73E-Y75E) that lie on the MAJIN dimerization interface. The authors demonstrate DNA binding of TERB2-MAJIN complexes that do not discriminate between double-stranded/single-stranded DNA or telomeric/non-telomeric DNA substrates. The DNA binding was shown to rely on basic patches in linker/unstructured regions in MAJIN and TERB2 as removal of these regions were associated with weaker binding. Dimerization of MAJIN also had a negative influence on DNA binding. DNA binding was further demonstrated with EM experiments that showed circular TERB2-MAJIN structures colocalize on DNA “strings”. The authors then build on the MAJIN-TERB2 core to characterize the MAJIN-TERB2-TERB1 complex using SEC-MALS and SAXS and ab initio modeling (in addition to their crystal structure) and find that MAJIN-TERB2-TERB1 exists 2:2:2, while TERB2-TERB1 exists 1:1. Using structured illumination microscopy they show that TERB2 is proximal to the chromosomal axis compared to MAJIN, which seems more distal from the chromosome axis. The authors then include TRF1 (the TRFH domain) into the complex to show that the TRF1-TERB1-TERB2 stoichiometry is 2:1:1, while TRF1:TERB1 stoichiometry is 2:1. They also build a model for this complex using SAXS ab initio methods. The authors then return to EMSAs to show that while TRF1 (TRFH) displaces MAJIN-TERB2 off DNA, the reverse is not true – full length TRF1 bound to DNA can recruit MAJIN-TERB2-TERB1. Finally TRF1 was shown to be more proximal to the chromosome axis than TERB1 or MAJIN while telomeric DNA (by FISH) was most distal. The greatest strengths of this work include the novel structure of the MAJIN-TERB2 interface, and the very impressive reconstitution and determination of stoichiometry of the various meiotic telomere complexes. However there are multiple disagreements between the current and published work that are pertinent to some of the major conclusions of the study. Furthermore the DNA binding studies need to be further strengthened to bolster some of the conclusions.

1. To make a strong argument about there being no discrimination for double vs. single stranded or telomeric vs. telomeric DNA, EMSA should be done under conditions where the [DNA] << Kd of the interaction. Also, as the authors are reporting a new protein-DNA interaction, it seems important to determine the absolute strength (Kd) of the interaction. For example, telomeric proteins like TRF1 and TRF2 are known to bind telomeric DNA with high affinity (nanomolar regime). Is the MAJIN-TERB2-DNA interaction weaker, and if so, how weak? If it indeed is in the micromolar regime – as suggested by crude inspection of the raw EMSA data, do the authors believe it is physiologically relevant?

8. We have repeated EMSA experiments using DNA at a concentration of 25 nM (per molecule), which we have analysed through densitometry and fitting to the Hill equation to calculate apparent affinities. Through this, we have determined that MAJIN_{Core}-TERB2_C interacts with DNA with apparent K_D values of 550 nM (random dsDNA), 610 nM (telomeric dsDNA), 900 nM (random ssDNA) and 510 nM (polydT ssDNA) (Figure 4d and Supplementary Figure 4f). We have further extended this analysis to the range of MAJIN-TERB2 constructs shown in Figure 4c, confirming our previous findings and revealing a higher affinity of 120 nM for MAJIN Δ TM-TERB2_C (containing the MAJIN unstructured C-terminus) (Figure 4d). As a control, we have used the same approach to measure the apparent K_D of TRF1 for dsDNA, which we find to be 100 nM (Supplementary Figure 5d), in agreement with previous reports (Hanaoka *et al* 2009 *Protein Sci* <https://doi.org/10.1110/ps.04983705>). Thus, MAJIN-TERB2 binds DNA with an affinity similar to that of TRF1 and is therefore likely to be physiologically relevant.

The original EMSAs that we presented utilised DNA substrates at a higher concentration of 0.3 μ M (per molecule), designed to provide clear visualisation of the DNA species formed to analyse binding mode (e.g. formation of larger molecular weight species at higher ratios) and to test for overt reduction in binding affinity. On the basis of the above analysis, this DNA concentration is below the apparent K_D of MAJIN_{Core}-TERB2_C (0.55 μ M), confirming its validity in demonstrating substantial reduction in affinity (such as for MAJIN_{Core}-TERB2_{C-Tr} and F73E Y75E). We think that the new analysis nicely complements the previous EMSA images as together they provide quantification of apparent K_D values with clear visualisation of DNA-binding modes and overt changes in affinity. Thus, we have added the new Hill equation fits and K_D determination alongside the EMSA images (Figure 4c-d) and have provided further explanation in the text (line numbers 180-184, 193-194, 195-196 and 375-376). Further, we recognise that having presented the DNA concentration in EMSAs as the per base pair concentration (25 μ M) may lead to confusion when considering their relation to affinities, so have replaced these with per molecule concentrations (0.3 μ M) (Figures 4d, 7c,f,g,i,j; Supplementary Figures 4e, 5c and 12d).

2. The authors show that unstructured/extended basic regions of both MAJIN and TERB2 are important for DNA binding. However neither of these regions are resolved in the crystal structure. As they propose a model where the DNA loops around MAJIN dimers along the two basic surfaces on the MAJIN lobes, the authors should validate this using mutations of basic residues in the MAJIN lobes. If the lobe mutants don't affect DNA binding, it would suggest that a) the DNA looping model needs to be revised, and b) all the DNA binding of MAJIN-TERB2 comes from basic linker/extended regions.

9. As suggested by the reviewer, we produced a MAJIN mutant in which the basic charge of the surface of the β -grasp domain was eliminated (K24M K26E R28E K31D R34E

R81D) (Supplementary Figure 3e-f). Surprisingly, this mutant inhibited dimerization of MAJIN-TERB2, instead producing a stable 1:1 complex (Figure 3a). Nevertheless, its complete elimination of DNA-binding, in comparison with the $\sim 1.8 \mu\text{M}$ binding affinity of the 1:1 complex formed by mutant F73E Y75E, confirms the importance of the MAJIN basic surface in DNA binding and validates our DNA looping model (Figure 4c-d). We have added updated these figures and added a description to the text (line numbers 200-205).

3. There is one major concern in the SEC-MALS experiment, which is in regard to the oligomeric status of TERB1 and TERB1-TERB2. The authors suggest that TERB1-TERB2 is a 1:1 complex. This goes against biochemical studies by Shibuya et al (Cell), which show that TERB1 in TERB1-C (which is TRFB+myb) as well as TERB1-C-TERB2 is homodimeric by gel filtration analysis, and that MBP-TERB1(TRFB) pulls down His-TERB1(TRFB) (see Supp fig 5 of that paper). There are at least two explanations for this discrepancy. Firstly, it is possible that the differences in techniques (SEC-MALS vs gel filtration alone or MBP pull down) are responsible for the incongruent observations. Secondly, the construct for TRFB used in the current study by Dunce et al is much shorter than the construct from Shibuya et al (starting at aa 561 as opposed to starting at aa 523). As stoichiometry determination is a major strength of the current study, it is important for the authors to resolve this difference. One direct route to resolve the disparate observations would be to perform SEC-MALS with TERB1(TRFB) as well as TERB1(TRFB)-TERB2 starting at TERB1 aa 523 rather than 561.

10. As the reviewer highlights, a previous study (Shibuya et al 2015 Cell <https://doi.org/10.1016/j.cell.2015.10.030>) reported that isolated TERB1-C (mouse residues 523-699) is a homo-dimer. This was based on its gel filtration elution profile and ability of MBP-TERB1-C to pull-down His-TERB1-C. However, examination of their data reveals that its gel filtration elution was between the 158 kDa and 670 kDa markers (Figure S5B), inconsistent with a simple dimer that would have a molecular weight of approximately 40 kDa (we would also argue that gel filtration elution points are an inherently inaccurate means of MW determination, hence our use of SEC-MALS to provide unambiguous masses). Further, they included no negative controls in their pull-down assay (Figure S5D), meaning that it is possible that His-TERB1-C simply aggregated on the amylose resin. Thus, the presented data are insufficient to substantiate their claim that TERB1-C is a homodimer. We further highlight that the authors also reported through similar analysis that the TRF1 dimer is disrupted upon binding to TERB1 to form a 1:1 homodimer. These unsubstantiated claims were subsequently disproven through crystal structures of TRF1-TERB1 showing the intact TRF1 dimer (Pendlebury et al 2017 NSMB <https://doi.org/10.1038/nsmb.3493>; Long et al 2017 NSMB <https://doi.org/10.1038/nsmb.3496>), and our finding that it forms a 2:1 complex.

To address this issue directly, we sought to establish the oligomeric state of our TERB1 TRFB construct (human residues 561-658) in isolation. We found that this protein

construct is unstable in absence of TERB2, and SEC-MALS analysis revealed its elution in the high molecular weight range of the column with molecular weight determination of 1.5-6 MDa (Supplementary Figure 6f). Interestingly, we were unable to establish complex formation upon incubation of this material with TRF1, indicating that these high molecular weight species reflect non-specific aggregates that cannot be broken down by TRF1. We further tested other isolated TERB1 constructs by SEC-MALS; whilst TERB1 TBM (human residues 642-658) formed a stable monomer (Supplementary Figure 13c), T2B (human residues 585-642) formed similar higher molecular weight aggregates (Supplementary Figure 6f). Thus, it appears that the TERB2-binding region of TERB1 TRFB is responsible for its aggregation, and is likely an *in vitro* artefact observed in absence of its binding partner (our ability to analyse TERB1 TRFB-TRF1 in absence of TERB2 was likely owing to the stabilising effect of TRF1 in preventing T2B-mediated aggregation). These findings lead to the suggestion that the TERB1-TERB2 complex may be constitutive within the cell.

The data presented by Shibuya *et al* can be readily explained by their TERB1-C construct undergoing similar non-specific aggregation as we observe for TERB1 TRFB. The gel filtration elution point they report is consistent with formation of high molecular weight aggregates, and their pull-down may either reflect binding within an aggregate or non-specific aggregation of His-TERB1 on the amylose resin.

The reviewer raised the possibility that the larger construct used in Shibuya *et al* may explain the difference. This appears unlikely as the additional residues involved show poor evolutionary conservation with no secondary structure prediction, and are likely to represent unstructured linkers. Nevertheless, we analysed a longer construct of human TERB1 522-665 (matching mouse sequence 523-699 used in Shibuya *et al*), and confirm by SEC-MALS that it forms a 1:1 complex with TERB2_N (see below).

We further highlight that our determination of TERB1-TERB2 as a 1:1 complex is supported by SEC-MALS analysis of its fusion protein (Supplementary 6f), of another TERB1-TERB2 construct (Supplementary Figure 6c), and our determination of 2:1:1 complex formation by TRF1-TERB1-TERB2 (Figure 6b).

We have added the additional figure panels described above (Supplementary Figures 6c,f and 13c), altered the figure legend (line numbers 1305-1306) and have added a description to the text (line numbers 221-223).

4. The authors claim that TRF1:TERB1 is 2:1 implying that only one site of a TRF1 dimer is bound by TERB1. This has been shown previously by Pendlebury *et al* (NSMB; figure 4) and is therefore not a new observation.

11. We agree that Pendlebury *et al* 2017 (NSMB <https://doi.org/10.1038/nsmb.3493>) demonstrated that only one of the two possible TERB1-binding sites of the TRF1 dimer is engaged in TERB1-binding (when using constructs similar to TRFB), but did not determine the stoichiometry of the complex. Our findings of a 2:1 complex explain, and are compatible with, the previous observation of Pendlebury *et al*; we have updated the discussion to clearly acknowledge this previous finding (line numbers 398-400).

5. In light of the model for the full DNA-TRF1---MAJIN-INM complex proposed here, it is surprising how the binding of TRFH to TERB1 inhibits binding of DNA to TERB2-MAJIN. As long linkers are separating the TRF1-TERB1 interface from the TERB2-MAJIN-DNA interface how do the authors invoke this inhibition is taking place? The binding data shown for TERB2-

TRF1 direct binding is very weak (Sup Figure 9E) and seems insufficient to explain the EMSA results. It is known that TRF1 has a very acidic N-terminus and so it is not surprising that it can associate simply by electrostatics to a positively charged surface such as the basic C-terminus of TERB2 (as shown in the supplement). However, it is difficult to argue based on this that the TERB2-TRF1 interaction is either specific or has implications for DNA-binding. Finally, if the authors propose that TRF1 pushes TERB2-MAJIN off DNA, that would suggest a very weak interaction of TERB2-MAJIN with DNA as it can be disrupted by addition of a protein (Kd determination with DNA as suggested above should be able to address this).

12. We agree that this is an unexpected result, but the inhibitory effect on DNA-binding by TRFH is robust. We have confirmed that the TERB2-TRF1 interaction is retained within the wider MAJIN-TERB2 complex, where it is again dependent on the TERB2 C-terminus (Supplementary Figure 12e).

It is important to remember that these molecules are tethered together through the TERB2 linker between the MAJIN-TERB2 and TERB2-TERB1-TRF1 complexes, and thus even if the TERB2-TRF1 interaction is relatively weak, it may be highly favoured owing to the increased local concentration that results from its physical tethering. The reviewer questions how this could be possible given the long linker of TERB2. We suggest that this linker is not necessary always fully extended, but could instead loop back to allow formation of a more compact ternary complex between MAJIN-TERB2-TERB1-TRF1. Indeed, we have illustrated this in Figure 8f-g, in which we show a looped linker prior to displacement, and then an extended linker post-displacement when the lack of TRF1 removes the driving force for ternary complex formation.

The reviewer suggests that the TERB2-TRF1 interaction may be mediated by the acidic N-terminus of TRF1; however, we would like to clarify that we demonstrated the interaction using the TRFH region of TRF1, which lacks the acidic N-terminus. Further, we demonstrate the specificity of this effect through the lack of DNA-binding inhibition when using a TERB1 construct (T2B) in which the TRF1-binding site is deleted (Figure 7g).

Finally, as described in response number 8, we have determined that the MAJIN-TERB2 core binds DNA with an affinity of 0.55 μ M (Figure 4d), indicating a substantial role of TRFH in being able to block this interaction. To fully resolve this issue would require crystallographic data of MAJIN-TERB2 bound to DNA, and MAJIN-TERB2 bound to TRFH, which have not yet proven possible. With this in mind, we acknowledge that we cannot provide a full molecular mechanism to explain this effect. Nevertheless, given the robustness of DNA-binding inhibition by TRFH, it appears worthy of reporting given that it helps to explain the ordered processes surrounding how telomeres are delivered to MAJIN-TERB2-TERB1 by TRF1.

We have added the additional pull-down data described above (Supplementary Figure 12e) and a description of this in the text (line number 293), and have added quantification of DNA-binding inhibition by TRFH (Figure 7h).

6. *Three different groups have shown that the phosphomimetic mutation of TERB1 T648 breaks the TERB1-TRF1 interaction (Shibuya et al; NCB, Pendlebury et al; NSMB; Long et al NSMB). The TERB1 constructs in these studies span the TBM and even full length TERB1, and employ different experimental methods (co-IP, fluorescence flow cytometry, yeast two hybrid, immunofluorescence in human cells, immunofluorescence in mouse spermatocytes). And in all cases the T648E/D mutation greatly reduces binding to TRF1. However, in the current study, this mutation doesn't affect TRF1 binding. One reason for this discrepancy could be the concentration at which the SEC studies were performed. Protein concentrations of 10 mg/ml were used (at least in SEC-MALS), which amounts to ~mM concentrations of TERB1(TRFB). As these concentrations are much higher than the reported Kd for the TERB1-TRF1 interaction (from the two NSMB studies), it seems highly likely that this is the reason why the authors do not see an effect of the T-to-E mutation. Direct Kd measurements need to be performed to substantiate the authors claim that there is no effect of the phosphomimetic mutation on the interaction. The TERB1 and TRF1 concentrations in vivo are probably much lower than mM, and hence the physiological TRF1-TERB1 interaction would be expected to still be sensitive to this mutation as has been already demonstrated by all three studies referenced above.*

13. As the reviewer describes, previous studies have implicated T648 phosphorylation in disruption of TRF1-binding by TERB1 through a variety of methods. Firstly, it has been demonstrated biochemically that phosphomimetic mutation T648E inhibits TRF1 binding of TERB1 TBM (Shibuya *et al* 2015 Cell; Pendlebury *et al* 2017 NSMB; Long *et al* 2017 NSMB). Alongside this, the phosphorylation event has been shown *in vivo*, with the T648E mutation being deleterious to telomere attachment. However, the ability of the mutation to block the TRF1 interaction of the wider TRFB region of TERB1 has never been tested biochemically. Further, the finding that mutation T648A fails to prevent TRF1 disruption *in vivo* (Shibuya *et al* 2015 Cell) indicates that this phosphorylation event is not sufficient to explain TRF1-TERB1 complex disruption.

We can now provide a simple explanation for the differing observations through additional analyses that we have performed on TRF1-TERB1 complexes. It was not possible to determine binding affinities between TERB1 and TRF1 as isolated TERB1 TRFB forms large molecular weight aggregates. Instead, we estimated relative binding affinities through analysing dissociation upon serial dilution by SEC-MALS (Supplementary Figure 13a-b). This revealed that the binding affinity of TERB1 TRFB for TRF1 is at least 100-fold greater than that of TERB1 TBM. TRFB formed a clear 2:1 complex when loaded at 5 μ M (corresponding to an analysis concentration of at most 1.25 μ M); lower concentrations were below the sensitivity of the instrument but the elution profile indicates retention of complex formation at a load concentration of 0.5 μ M (analysis at 0.125 μ M). In contrast, TBM formed only a partial complex when

loaded at 500 μM (analysis at 125 μM). In agreement with previous studies, the TBM interaction was largely inhibited by T648E mutation (Supplementary Figures 10b and 13d). However, the much stronger interaction of TRFB was only weakened by the T648E mutation, with its SEC-MALS dissociation pattern indicating a reduction in binding affinity by approximately 10-fold relative to wild type protein (Supplementary Figure 13a-b). Importantly, its 2:1 complex formation was retained when loaded at 50 μM (analysis at 12.5 μM) and dissociation to free TRF1 was only clearly observed when loaded at 0.5 μM (analysis at 0.125 μM), and thus the binding affinity of TRFB T648E is greater than 10-fold stronger than wild type TBM (Supplementary Figure 13a-c). Thus, T648 phosphorylation does weaken TRF1 binding of TERB1, but the effect is much more apparent (and seemingly absolute) in the TBM peptide in which the affinity is already greatly reduced.

We further attempted to disrupt the TRF1-TERB1 complex through phosphorylation by CDK1-CyclinB (Supplementary Figure 13e-h). This was successful for TERB1 TBM, in which we detected substantial *in vitro* phosphorylation and complex dissociation (Supplementary Figure 13e,g,h). However, TERB1 TRFB showed a relative resistance to phosphorylation (despite being exposed to a 10-fold high molar ratio of enzyme), indicating that the structure of the complex hinders access of CDK1-CyclinB to the target site, with retention of the 2:1 complex (Supplementary Figure 13f-h). Thus CDK1-CyclinB phosphorylation of T648, and other sites within TRF1 and TERB1, appears insufficient to achieve disruption of the TRF1-TERB1 complex when using the high affinity TRFB sequence. We suggest that other cellular events must collaborate with T648 phosphorylation to achieve TRF1 disruption. Moreover, an initial event may loosen the complex, permitting access of CDK1-CyclinB, with resultant T648 phosphorylation completing disruption of the complex.

The reviewer raises the question of the concentration at which SEC-MALS experiments were performed. Firstly, we would like to clarify that they were performed at >1 mg/ml, whereas SEC-SAXS experiments were performed at >10 mg/ml (to maximise the SAXS scattering signal). These concentrations correspond to TRF1-TERB1 molar concentrations of 15 μM and 150 μM , respectively. As there are SEC methods, samples experience 5-10-fold dilution before analysis by MALS or SAXS, so analysis was performed at less than 3 μM and 30 μM , respectively. Secondly, our original description of stability of the TRFB T648E mutant complexes was based on their perfect co-purification through multiple chromatography steps, including affinity, ion exchange and size-exclusion chromatography, indicating stability in a variety of chemical conditions and over a range of concentrations. Nevertheless, the additional SEC-MALS serial dilution analysis described above directly addresses this issue and provides clarity regarding the ability of T648 phosphorylation to disrupt TERB1 TBM and TRFB constructs.

We have added the additional experimental data described above (Supplementary Figures 10b and 13), along with figure legends (line numbers 1261-1263, 1311-1341), and have added a description of this to the text (line numbers 309-325 and 412-418).

Further, on the basis of our findings of weak binding between TERB1 TBM and TRF1, we note the disparity in its reported binding affinities (5.6 μ M and 75 nM). The former study (Long et al 2017 NSMB) describes a binding affinity in keeping with our observations. However, the latter study (Pendlebury et al 2017 NSMB) describes a much stronger binding affinity. This can be explained by their use of a GST-fusion peptide, in which the GST-induced dimerization of TERB1 TBM peptides may have enhanced their apparent affinity through the cooperativity resulting from simultaneous binding to both sites of the TRF1 dimer. We speculate that the high affinity of TERB1 TRFB may result from a similar two-site cooperative binding event in which the N-terminus (561-585) binds to a second binding site within the TRF1 dimer. Thus, the use of a GST dimer may have mimicked the binding mode of the wider TRFB region of TERB1. We have added a discussion of this to the text (line numbers 397-406).

7. In all of the SIM experiments done in mouse spermatocytes, the sub-stage of pachytene is not indicated. The colocalization of TERB1-TERB2-MAJIN with shelterin (including TRF1) and telomeric DNA depends on the stage in pachytene (Shibuya et al; Cell). In early pachytene all these proteins colocalize, while later in pachytene the signals of these complexes separate. In the SIM results shown in the current study the signals of TRF1 seem partially separated (and chromosome axis proximal) from that of TERB1-TERB2-MAJIN. Does this mean that all of the images captured represent late pachytene? If so, a marker for late pachytene should be included. If the images have not been sorted for a particular stage in pachytene, it is surprising that TRF1 has “detached” from the telomeric signal so early; this would not follow the imaging studies from the Watanabe lab on cap exchange. On a related note, some of the profile changes in the SIM experiments seem subtle and therefore inclusion of statistical criteria (standard deviations, p-value, etc) seems important.

14. All SIM experiments were performed in late pachytene. Prophase sub-staging was performed according to criteria previously defined in the literature relating to the characteristic changes of the XY body during pachynema (M.J. Moses, Animal Models in Human Reproduction. Raven Press, New York, 169:190, 1980). In particular, the extension of synapsis decreases from early to late pachynema, with X and Y chromosomes adopting a characteristic end-to-end association in late pachynema. We have added a description of this to the methods (line numbers 675-695) and have indicated the stage in the text (line numbers 237, 240-241, 332-335 and 338-340) and figures (Figures 5f and 8c-e; Supplementary Figures 8, 9, 14 and 15).

In accordance with response number 7, we have also added additional analyses of zygotene spermatocyte chromosomes (Figures. 5e and 8a; Supplementary Figures 8a and 14a-b).

Please see response number 18 regarding quantification of the SIM images.

8. *The location of the MAJIN basic patch linkers towards the membrane suggests a role for them associating with the inner nuclear membrane containing phospholipids. The DNA binding activity could be an artifact of the absence of any membrane component in the experiment. Although it is difficult to adapt the EMSA to address this, the authors could perform INM-tethering experiments as performed previously by the Lei and Watanabe labs with BP1/BP2 mutation constructs to ask if these disrupt membrane binding.*

15. The reviewer suggests that the basic patches of the MAJIN linker may be responsible for binding to the inner nuclear rather than to DNA. However, we point out that these basic patches are clustered next to the MAJIN-TERB2 globular structure rather than being proximal to the membrane (Figure 4a), and so when the linkers are extended they may be up to 40 nm away from the inner nuclear membrane. As the reviewer indicates, it is not technically possible to incorporate inner nuclear membrane to EMSA DNA-binding assays. Nevertheless, the Watanabe lab has already performed INM-tethering experiments demonstrating that MAJIN retains its ability to localise to the INM upon mutation of its linker basic patches (Shibuya *et al* 2015 Cell, Figure 4G), supporting our model of their role in DNA-binding.

Minor points:

1. *The method of expression and purification, and the quality of the purified full length TRF1 protein should be included as very few biochemical studies have been performed with purified full-length TRF1 presumably because it is a difficult protein to study biochemically.*

16. We have added further details of the purification to the methods (line numbers 480-489), an SDS-PAGE summary of the TRF1 purification (Supplementary Figure 5a), and our validation of its dimeric structure through SEC-MALS analysis (Supplementary Figure 5b).

2. *It is not clear how the SAXS data are used in the CORAL or ab initio modeled structures. There doesn't seem to be any SAXS envelope suggestive of these linkers.*

17. The SAXS CORAL models were generated through fitting directly to SAXS scattering data rather than to SAXS *ab initio* envelopes, MONSA analysis generated multi-phase *ab initio* models directly from SAXS scattering data, and other *ab initio* envelopes were generated from SAXS real space distributions. We have added further in the figure legends (line numbers 993 and 1035-1038).

3. *The number of replicates “n” is not indicated in the quantitations in figure 8 b, c, d. If these are for representative telomeres, then replicates should be quantified so that at least tens of foci are quantified.*

18. Quantification is displayed for frontal orientations where it is possible to align multiple telomere images on the basis of the SYCP3 chromosome axis signal. In these cases, we demonstrate frontal co-localisation of TERB1-TERB2, MAJIN-TERB2 and TERB1-TRF1, but separation of MAJIN-TRF1 ($n > 20$) (Figures 5f and 8c-d). However, similar quantification is not possible for other orientations where clear but irregular differences emerge in the distributions of MAJIN-TERB2 and TERB1-TRF1. The irregular nature of these structures by definition precludes accurate alignment of multiple telomere images, and an attempt to average such irregular distributions would simply cancel out and provide meaningless plots. Nevertheless, the individual images we present are representative of at least 20 observations of each antibody combination. We have extended our previous analysis and now include multiple orientations of different telomere ends for each antibody combination (Figures 5e-g and 8a-e; Supplementary Figures 9a-d and 15d-e). We have further included wide-field images (Supplementary Figures 8a-f, 14a-f and 15a-c,f), from which numerous instances of each orientation are visible, which support findings presented in the main figures.

4. *A membrane stain (if this is feasible) would be very useful in the SIM experiments so that one can calibrate the distance from both the chromosomal and membrane directions in each panel.*

19. It is unfortunately not technically possible to incorporate a membrane stain into our SIM experiments on spread meiotic chromosomes. This is because we are already doing triple immunofluorescence using rabbit, guinea pig and mouse as primary antibodies for MAJIN/TERB2/TERB1 proteins, TRF1 and SYCP3 (meiotic chromosome axis marker), and antibodies for the nuclear membrane produced are not currently available in different hosts.

5. *TIN1 should be changed to TIN2 in line 88.*

20. We agree and have corrected this (line number 89).

Reviewer #3 (Remarks to the Author):

The authors present an impressively extensive study on the MAJIN-TERB2-TERB1 complex. Utilizing X-ray crystallography combined with solution scattering and biochemical techniques, they seek to solve the structure of this complex in solution and give insights into the telomere attachment mechanism and subsequent manipulation.

This reviewer has experience primarily in the small-angle x-ray scattering (SAXS) aspect of the study and therefore will primarily comment on this aspect of the study. In short, while it is possible that the SAXS data are sufficient, the data are not presented with enough clarity and controls to ensure that the reported conclusions are accurate and represented by the data. In this review's opinion, the paper could be published only if these concerns are addressed through additional analysis being presented.

Overarching comments:

1. There should be a concentration series to control for common recurring issues in SAXS.

a. The first is to control for aggregation. Aggregation effects have been attempted to be controlled through SEC-SAXS methods, but these are insufficient. They do not rule out effects due to time (the molecules spend a significant amount of time on a molecular time scale outside of the separation column) or radiation-induced aggregation effects. The currently reported flow rate (0.5ml/min or ~8ul/s) means that 8ul of sample are being exposed to 1 second of up to 10^{12} photons per second (as the authors do not report a beam strength, the reviewer assumes the full beam available at B21 of the Diamond Light Source was used). This flux is more than sufficient to cause radiation damage in certain proteins.

b. The second possible issue is interparticle interference. Due to running at high concentrations the experiments in this study are especially likely to encounter these effects. Both interparticle interference and aggregation can have large effects at the lowest q values, which affect the measurement of the largest dimensions of the sample.

21. We agree that aggregation and interparticle interference are important concerns in SAXS experiments, and we explain how we have mitigated these effects as follows:

- a. As the reviewer points out, all analyses have been performed by SEC-SAXS as a means of eliminating pre-formed aggregates. To mitigate the possibility of aggregates forming during the flow between SEC elution and the test cell, we tested identical samples (in terms of prep, volume and concentration), using the same column type, volume, flow rate and buffer, with a similar flow system by SEC-MALS. In all cases, we obtained reliable molecular weight determination, with mostly flat molecular weight plots across peaks, other than some downslopes for samples containing minor degradation.
- b. In response to the possibility of radiation-induced aggregation, we have consulted with the principal beamline scientist of B21 at Diamond Light Source, who has provided the following important information and analysis regarding

the beamline set-up. The beam size is 1x1 mm and the capillary has a diameter of 1.5 mm, so the illuminated volume is approximately 1.8 μ l. Our experiments were run at 0.5 ml/min = 8 μ l/s, corresponding to protein exposure to the beam path for only 0.225 s. This can be considered a sample exposure of approximately 20% of the full beam, which is in the range of 10^{11} photons per second. This represents a relatively low exposure as analysis of BSA (without flow) with a full beam has shown radiation damage after 2.5-5s exposure in PBS and much longer for buffers containing a carbon or nitrate source (our data were collected in Tris). On this basis, it is estimated that we would have to perform runs at less than one tenth of our current flow rate (< 0.8 μ l/s) to expose the sample to conditions that generate radiation damage for BSA. We have added details regarding illuminated volume and proteins/sample exposure time to the methods (line numbers 535-537).

- c. Interparticle interference can be diagnosed by analysing the R_g across the peak as it results in a sharp decrease (under-estimate) in R_g as the sample concentration increases and exceeds the threshold, and then a return to the previous value as the concentration falls below the threshold after the peak. In all cases, we performed per-frame analysis of R_g across elution peaks and confirmed that the R_g remained constant throughout (or reduced slightly owing to minor degradation). Further, the SEC column we used results in 5-10 fold dilution of samples, meaning that whilst proteins were loaded at >10 mg/ml (typically no more than 15-20 mg/ml), SAXS analysis was performed at approximately 1-4 mg/ml.
- d. The reviewer suggests the use of a concentration series as a further means to control for aggregation and interparticle interference. However, as the reviewer subsequently points out (query number 2), concentration series are difficult on SEC-SAXS. The reason for this is that each run takes approximately one hour (24 ml column at 0.5 ml/min), so it is not feasible to obtain sufficient synchrotron beamtime to analyse a concentration series of a substantial number of samples. Further, we find that protein concentrations in the experimental cell of at least 0.5 mg/ml are necessary to obtain a sufficient signal:noise ratio (with significant improvement >1.5 mg/ml), which given 5-10 fold dilution by SEC, requires samples to be loaded at concentrations of >5 mg/ml. Thus, it is necessary to load samples at high concentration to obtain interpretable scattering curves and substantial dilution is unlikely to provide meaningful data. Nevertheless, for each SAXS analysis we present, we have typically collected several datasets on different occasions, from different protein preps, at slightly different concentrations, and in many cases of slightly different construct boundaries. In all cases, we obtained similar results in terms of scattering curves, R_g values, real space $P(r)$ distributions, ab initio models and rigid body/linker models (or slight differences that are explained by alteration in construct boundaries).

2. Concentration series are harder on SEC-SAXS systems. Barring the possibility to do these, extensive Guinier analyses should be completed. However, at no point in the analysis do the authors present their Guinier regimes or their R_g values. These should both be presented at least in the supplementary information. Additionally, for all $P(r)$ determinations, the authors should ensure that the R_g calculations roughly match the Guinier analysis.

22. We had performed detailed Guinier analyses of our data but had excluded these for brevity. In light of the reviewer's request, we have provided Guinier analyses highlighting the Guinier regions and linear fits, along with R_g values, for all samples (Supplementary Figures 4a-b, 7a-b and 11e-f). We also confirm that the real-space $P(r)$ R_g values match the Guinier R_g values for all samples, and have added these details to the legends of Figure 3c and Supplementary Figures 7c-d and 11g (line numbers 988-989, 1234-1236 and 1289-1291).

3. A constant issue with SEC-SAXS is buffer subtraction, due to the often long times between sample and background buffer measurements. The authors should explain how careful background subtraction was achieved. Additionally, as mentioned above, a transparent Guinier analysis would allow the reader to evaluate the quality of the buffer subtraction.

To see the possible pitfalls of SEC-SAXS buffer subtraction and proposed fixes, please see the work of Malaby et al.[1]

23. Background subtraction was performed using buffer collected from the SEC run of the sample either before or after the protein elution (depending on the position of the peak relative to the void/buffer exchange and the presence of any additional species). The position of the background range was directed by flat regions of the integral of the ratio of each frame (plotted using ScÅtter 3.1), and background frames were inspected individually to ensure consistency. Further, subtraction was often performed with different regions of the SEC elution and resultant scattering curves were compared. Guinier analysis was performed and data were checked for the presence of any non-random noise in the low- q range. We have added a statement regarding background subtraction to the methods (line numbers 537-539) and have added Guinier analyses to the Supplementary Figures (as described in response number 22).

4. D_{max} is an important and difficult parameter to determine when calculating $P(r)$. There are numerous instances in the manuscript where the utilized D_{max} values are suspect. $P(r)$ profiles should normally approach zero naturally at D_{max} in a concave manner.[2] There are frequent times in the manuscript where the authors have likely chosen an artificially small D_{max} value.

*While this seems like a small thing, the D_{max} value has a strong effect on the $P(r)$ profile and the consequent *ab initio* reconstructions.*

24. We agree that D_{max} is a difficult parameter to fit, and on reflection we also agree that in some cases it had been slightly truncated and lacked the natural approach to zero with concavity. On this basis, we have re-processed a number of SEC-SAXS datasets and repeated *ab initio* modelling and subsequent rigid-body/linker modelling. We have updated the figures and figure legends accordingly (Figures 3c-d and 5c-d; Supplementary Figures 7c-e and 11g; line numbers 987-991, 1035-1041, 1232-1238 and 1287-1289), as described subsequently in response to specific comments. Importantly, the updated processing has resulted in similar and in some cases slightly improved *ab initio* models, and our findings remain unaffected.

5. *For all SAXS profiles, the authors should include y-axis units. If the units are “arbitrary” they should be indicated as such.*

25. The y-axis units are arbitrary and we have added the indication (a.u.) to all instances (Figures 3b, 5d and 6c; Supplementary Figures 4a-b, 7a-b and 11e-f).

6. *There are two excellent articles the author suggests for handling of SAXS data and D_{max} determination by Skou et al.[3] and the aforementioned Jaques et al.[2]*

26. We thank the reviewer for the suggestion; accordingly, we have reviewed the articles and have taken them into account when re-processing data (as described in response number 24).

Specific comments:

1. *Figure 3b: As said in the general comments, one would need to see a Guinier analysis to determine whether or not the experimental data is adequate. In particular, the reviewer is worried about the very low q data for the top SAXS profile.*

27. We have added Guinier analyses for these datasets to Supplementary Figure 4a-b. The scattering data presented in Figure 3b (and in other panels) include all collected experimental data, including noisy data in the very low- q range close to the beamstop that had been excluded from $P(r)$ distribution calculation on the basis of Guinier analysis. For clarity, we have updated all panels such that data below the Guinier region are plotted in grey to allow ease of demarcation (Figures 3b, 5d and 6c) and have added a description of this to figure legends (line numbers 985-986, 1041 and 1070-1071).

2. Figure 3c: The dotted $P(r)$ profile (MAJINCore-TERB2C) likely has an artificially low D_{max} which is what leads to the “shoulder” around 100 angstroms. The solid, black line (MAJINCoreTr-TERB2C-Tr) also may have a slightly artificially truncated D_{max} .

28. As described in response number 24, we have re-processed the data for both constructs and have updated Figure 3c and its figure legend (line numbers 987-989) with $P(r)$ distributions showing slightly longer D_{max} values. We have further repeated the *ab initio* modelling on MAJIN_{Core-Tr}-TERB2_{C-Tr} utilising the new $P(r)$ distribution, and have updated the model (Figure 3d) and its figure legend (line number 991). The resultant model shows similar features to that previously presented and our conclusions remain unchanged.

3. Figure 4d: The reviewer was confused by the bottom-right figure. Should the reviewer be seeing anything in this picture? If not, it would be helpful to include a statement explaining what it seeks to display.

29. This EM panel shows the lack of assembly (no apparent structures of sufficient size for EM visualisation) of MAJIN_{Core}-TERB2_C F73E Y75E in comparison with wild type. We have clarified this point in the figure legend (line numbers 1019-1021).

4. Figure 5d: The top SAXS profile (TERB2-TERB1) does not go to the same low q value as all of the other data. Is there a reason for this? The reviewer is concerned that there were problems with this low- q data and this is why it was not included.

30. We have updated Figure 5d to show the TERB2_N-TERB1_{T2B} data across the full q -range, with low- q data below the Guinier region shown in grey (as described in response number 27), with Guinier analysis shown in Supplementary Figure 7b).

5. Figure 5d: All other Crysol fits in the manuscript go to zero. The “CORAL model” theory curve (gray curve) does not go to $q=0$. It is also truncated at high- q (see, for comparison, see the profile directly above it in red.) These points of the fit should be included so that the reader can accurately evaluate the model.

31. We have repeated the CORAL modelling (Figure 5c) utilising the updated *ab initio* model of TERB2_N-TERB1_{T2B} (described in response number 32). We have updated the fit shown in Figure 5d such that it extends across the full q -range.

6. SupplementalFigure5e: Again, these D_{max} values seem problematic. Especially worrisome is the TERB2_N-TERB1_{T2B} profile that is subsequently used to create the *ab initio* model in 5f.

32. As described in response number 24, we have re-processed the data and have presented the new $P(r)$ distributions in new Supplementary Figure 7c-d. We repeated TERB2_N-TERB1_{T2B} *ab initio* modelling utilising the new $P(r)$ distribution (Supplementary Figure 7e) and updated the fit in Figure 5d and its figure legend (line number 1040). We further repeated CORAL modelling of Figure 5c utilising the updated *ab initio* models (as described in response number 31). The new *ab initio* and CORAL models show the same features as previously, and our findings remain unchanged.

7. Figure 6c: *There seem to be subtraction issues for the TERB2-TERB1 and TRF1-TERB1-TERB2 scattering profiles. In the former, the lowest q data seems to increase; in the latter, the lowest q data rapidly decreases. Again, a Guinier analysis would make these issues clearer and possibly allow the authors to obtain a better analysis.*

33. The lowest q data are below the Guinier regions and likely reflect noise close to the beamstop; we have updated Figure 6c to show these ranges in grey (as described in response number 27), and have added Guinier analyses for all datasets (Supplementary Figures 11e-f). In all cases, we have checked data processing by utilising different regions for buffer subtraction and obtain similar scattering profiles.

8. Figure 6d: *The authors should show at least one rotation of the final structure to evaluate whether the crystal structure fits into the ab initio model in the third (into/out of page) dimension.*

34. We confirm the good fit of the crystal structure through providing a rotation of the model as requested (Figure 6d).

9. Supplemental Figure 8e: *There seems to be an issue with two of the Dmax values in the P(r) profile. The TRF1-TERB1-TERB2 profile has a well-determined Dmax value as indicated by the slope of the profile whilst approaching Dmax. The other two have more abrupt terminations. The reviewer also wonders why the TERB2_N-TERB1_{T2B} P(r) profile matching the scattering profile in Figure 6c was not included?*

35. As described in response number 24, we have re-processed the data and have presented new $P(r)$ distributions in new Supplementary Figure 11g. We have added in the $P(r)$ distribution for TERB2_N-TERB1_{T2B} (Supplementary Figure 11g) and have edited the figure legend accordingly (line numbers 1287-1291).

Reviewers' comments:

Reviewer #1 (Remarks to the Author):

The authors have satisfactory answer all concerns that this reviewer raised.
In my opinion the manuscript is ready for publication.

Reviewer #2 (Remarks to the Author):

The authors have done an outstanding job of addressing Reviewers' concerns. In this Reviewer's opinion, this study is ready for publication in a high-impact journal such as Nature Communications.

Reviewer #3 (Remarks to the Author):

The changes that have been made to the manuscript are significant and respond to many of my previous concerns. In particular, the D_{max} values have been much more carefully addressed in all of the $P(r)$ analyses. However, there are still a few significant issues that I believe must be addressed:

1. The added Guinier analyses greatly enhance the ability for the reader to assess the quality of the SEC-SAXS data. However, some of the issues these analyses present show exactly why these data are so crucial. In many of the Guinier analyses, there is noticeable "smiling" of the low- q SAXS data (increase in low- q signal) that indicates likely aggregation in small amounts or buffer subtraction issues. In particular, figures S4a (top trace), S4b, S7a, or S11e (top trace). These can be compared to Figures S4a (bottom trace), S7b, or S11e (bottom trace). While it is difficult to see, partially because the range of interest is often only a small part of the plot, it seems clear that there is some non-linearity to these traces. At the very least this should be addressed in the text.
2. It is possible that the actual R_g regimes are linear, but it is difficult to assess due to the scales used. I would recommend plotting only the Guinier regimes of each trace so that the reader can fully determine the linearity of the signal.
3. The issues in the Guinier regimes leads to some (possibly) problematic results. In particular, while the text claims that the Core (Untruncated) and Truncated versions of the MAJIN-TERB2 complex are nearly identical, the R_g values differ by 20-25%. This is almost as large of an R_g difference as between the Untruncated and DeltaTM structures. This is possibly due to the slight aggregation in the Untruncated complex (top trace, S4a) compared to the Truncated (bottom trace, S4b) which shows no upturn in the low- q regime. Another possibility is that despite having similar crystal structures, the solution structures of these two complexes are quite different. This discrepancy would be significant as the Truncated and DeltaTM signals are used for the subsequent CORAL analyses. It should also be noted that this also seen in the D_{max} values, where the D_{max} for the Untruncated structure is much larger than the Truncated structure. This should be addressed directly with an explanation as to why these complexes are so different in solution when they appear identical in crystal form. It also begs the question of how this affects the proceeding analysis where the truncated version is used to create the SAXS scattering envelope. It is possible that the long, extended tails shown in the CORAL model of Figure 3 would be much smaller if the Untruncated version of the MAJIN-TERB2 complex was used.
4. In at least two of the supplementary figures (Figure S4a and Figure S11e) the x-axis tick mark labels are inconsistent. The first tick is 0.0001 while the second is 0.002 (missing a zero). This reviewer strongly suggests allowing the plotting programs to label the axes rather than labeling axes by hand.

Response to reviewers' comments
'Structural basis of meiotic telomere attachment to the nuclear envelope by
MAJIN-TERB2-TERB1'

We are pleased that all three reviewers recognise the changes that we have made to the manuscript. Here, we provide responses to the remaining issues raised by reviewer 3, and outline how we have revised the manuscript accordingly.

Reviewer #3 (Remarks to the Author):

The changes that have been made to the manuscript are significant and respond to many of my previous concerns. In particular, the D_{max} values have been much more carefully addressed in all of the $P(r)$ analyses. However, there are still a few significant issues that I believe must be addressed:

1. The added Guinier analyses greatly enhance the ability for the reader to assess the quality of the SEC-SAXS data. However, some of the issues these analyses present show exactly why these data are so crucial. In many of the Guinier analyses, there is noticeable "smiling" of the low- q SAXS data (increase in low- q signal) that indicates likely aggregation in small amounts or buffer subtraction issues. In particular, figures S4a (top trace), S4b, S7a, or S11e (top trace). These can be compared to Figures S4a (bottom trace), S7b, or S11e (bottom trace). While it is difficult to see, partially because the range of interest is often only a small part of the plot, it seems clear that there is some non-linearity to these traces. At the very least this should be addressed in the text.

1. To clarify this issue, we have re-plotted all Guinier analyses to focus on their Guinier regions (described in response number 2). This shows that all Guinier regions are essentially linear, within the realm of experimental error. The plots highlighted by the reviewer as showing potential signs of smiling do not appear to demonstrate any general trends on close examination, but likely had this appearance owing to the presence of the wider q range. Instead, they show fluctuations around the linear fit consistent with experimental error, with noise in the very low- q data below the Guinier region owing to the proximity of the beam stop. Overall, whilst we can never rule out the presence of small amounts of aggregation, the Guinier analyses suggest that this would be relatively minor and unlikely to affect data interpretation. We have added a statement to the methods indicating that the Guinier regions are essentially linear but, in some cases, show fluctuations depending on the signal:noise ratio of the particular construct, and that subsequent analysis was performed excluding data below the minimum q -value of the Guinier region (line numbers 542-546). The updated Guinier plots will further aid the reader to assess the linearity directly.

2. *It is possible that the actual R_g regimes are linear, but it is difficult to assess due to the scales used. I would recommend plotting only the Guinier regimes of each trace so that the reader can fully determine the linearity of the signal.*

2. We have re-plotted all Guinier analyses with scales that focus on the Guinier regions (Supplementary Figures 4a-c, 7a-b and 11e-g). We have also included linear fits in red to highlight the linearity of the Guinier regions. We have updated figure legends accordingly (line numbers 1184-1185, 1236-1237 and 1291-1292).

3. *The issues in the Guinier regimes leads to some (possibly) problematic results. In particular, while the text claims that the Core (Untruncated) and Truncated versions of the MAJIN-TERB2 complex are nearly identical, the R_g values differ by 20-25%. This is almost as large of an R_g difference as between the Untruncated and DeltaTM structures. This is possibly due to the slight aggregation in the Untruncated complex (top trace, S4a) compared to the Truncated (bottom trace, S4b) which shows no upturn in the low- q regime. Another possibility is that despite having similar crystal structures, the solution structures of these two complexes are quite different. This discrepancy would be significant as the Truncated and DeltaTM signals are used for the subsequent CORAL analyses. It should also be noted that this also seen in the D_{max} values, where the D_{max} for the Untruncated structure is much larger than the Truncated structure. This should be addressed directly with an explanation as to why these complexes are so different in solution when they appear identical in crystal form. It also begs the question of how this affects the proceeding analysis where the truncated version is used to create the SAXS scattering envelope. It is possible that the long, extended tails shown in the CORAL model of Figure 3 would be much smaller if the Untruncated version of the MAJIN-TERB2 complex was used.*

3. The statement that the core and truncated versions of the complex are essentially the same relates to their crystal structures, which overlay with an r.m.s. deviation of 1.61 Å. The additional residues of the core complex were not visible in the electron density, indicating that they are largely unstructured and do not contribute to the globular fold. It was on this basis that we designed the truncated construct, in which unstructured C-termini of MAJIN and TERB2 were removed, producing crystals that diffracted to higher resolution. We agree that SAXS data show differences between constructs, with higher R_g and D_{max} for the core complex. These differences, in particular the long tail of the $P(r)$ distribution, are consistent with the same globular fold but the additional presence of unstructured/flexible MAJIN and TERB2 C-terminal sequences in the untruncated core complex. We have added a statement clarifying this point to the manuscript (line numbers 140-141).

It is important to point out that the SAXS data in question (core untruncated complex) were not used for any subsequent analysis other than in presenting its $P(r)$ distribution alongside other constructs for completeness (Figure 3c). SAXS data on the truncated complex were used to generate the ab initio envelope that closely matches the crystal structures solved for both truncated and untruncated complexes. The ability of the

untruncated complex to crystallise in a distinct crystal form, with a structure matching that of the crystalline and solution states of the truncated complex, strongly supports that both complexes contain the same globular structure, differing only in additional unstructured sequence.

The reviewer raises the question of whether the difference in SAXS profiles of the core and truncated complexes may affect the CORAL analysis of the DTM complex shown in Figure 3e. We also appreciated that the CORAL analysis may be confounded by the presence of unstructured ends on multiple components as it would present an ambiguity as to which component is responsible for elongation. For this reason, we performed the analysis using a DTM complex containing the truncated TERB2 sequence, ensuring that the only difference between the DTM complex analysed and the truncated complex (and thereby the crystal structure) is the presence of long MAJIN C-termini. Thus, the only modelled components of the CORAL DTM analysis are the MAJIN C-termini. In contrast, analysis using a complex containing TERB2 core would require modelling of both MAJIN and TERB2 C-termini, providing additional ambiguity.

4. In at least two of the supplementary figures (Figure S4a and Figure S11e) the x-axis tick mark labels are inconsistent. The first tick is 0.0001 while the second is 0.002 (missing a zero). This reviewer strongly suggests allowing the plotting programs to label the axes rather than labeling axes by hand.

4. We thank the reviewer for pointing out this error, which has been corrected as part of re-plotting the Guinier analyses (described in response number 2).